# Water restrictions under climate change: a Rhone-Mediterranean perspective combining 'bottom up' and 'top-down' approaches

Eric SAUQUET[1], Bastien RICHARD[1,2], Alexandre DEVERS[1], Christel PRUDHOMME[3,4,5]

*Correspondance to*: E. Sauquet (eric.sauquet@irstea.fr)

[1] Irstea, UR Riverly, 5 rue de la Doua CS20244, 69625 Villeurbanne cedex, France

[2] Irstea, UMR G-EAU, Water resource management, Actors and Uses Joint Research Unit, Campus Agropolis - 361 rue Jean-François Breton – BP 5095, 34196 Montpellier Cedex 5, France

[3] European Centre for Medium-Range Weather Forecasts, Reading, UK

[4] Department of Geography, Loughborough University, Loughborough, LE11 3TU, UK

[5] NERC Centre for Ecology & Hydrology, Maclean Building, Benson Lane, Crowmarsh Gifford, Wallingford, Oxon, OX10 8BB, UK

**Abstract** Drought management plans (DMPs) require an overview of future climate conditions for ensuring long-term relevance of existing decision-making processes. To that end, impact studies are expected to best reproduce decision-making needs linked with catchment intrinsic sensitivity to climate change. The objective of this study is to apply a risk-based approach through sensitivity, exposure and performance assessments to identify where and when, due to climate change, access to surface water constrained by legally-binding water restrictions may question agricultural activities. After inspection of legally-binding water restrictions (WR) from the DMPs in the Rhône-Méditerranée (RM) district, a framework to derive WR durations was developed based on harmonized low-flow indicators. Whilst the framework could not perfectly reproduce all WR ordered by state services, as deviations from socio-political factors could not be included, it enabled to identify most WRs under current baseline, and to quantify the sensitivity of WR duration to a wide range of perturbed climates for 106 catchments. Four classes of responses were found across the RM district. The information provided by the national system of compensation to farmers during the 2011 drought was used to define a critical threshold of acceptable WR, related to the current activities over the RM district. The study finally concluded that catchments in mountainous areas, highly sensitive to temperature changes, are also the most predisposed to future restrictions under projected climate changes considering current DMPs, whilst catchments around the Mediterranean Sea were found mainly sensitive to precipitation changes and irrigation use was less vulnerable to projected climatic changes. The tools developed

enable a rapid assessment of the effectiveness of current DMPs under climate change, and can be used to prioritize
review of the plans for those most vulnerable basins.
**Keywords** Climate change; drought management plan; low-flow; France; scenario-neutral approach; response
surface; vulnerability; water restriction.

## 33  1 Introduction

The Mediterranean region is known as one of the "hot spots" of global change (Giorgi 2006; Paeth *et al*. 2017)
where environmental and socio-economic impacts of climate change and human activities are likely to be very
pronounced. The intensity of the changes is still uncertain, however, climate models agree on significant future
increase in frequency and intensity of meteorological, agricultural and hydrological droughts in Southern Europe
(Jiménez Cisneros *et al*. 2014; Touma *et al*. 2015), with climate change likely to exacerbate the variability of
climate with regional feedbacks affecting Mediterranean-climate catchments (Kondolf *et al.* 2013). Facing more
severe low-flows and significant losses of snowpack, southeastern France will be subject to substantial alterations
of water availability: Chauveau *et al.* (2013) have shown a potential increase in low-flow severity by the 2050's
with a decrease in low-flow statistics to 50% for the Rhône River near its outlet. Andrew and Sauquet (2017) have
reported that global change will most likely result in a decrease in water resources and an increase both in pressure
on water resources and in occurrence of periods of water limitation within the Durance River basin, one of the
major water tower of southeastern France. In addition, Sauquet *et al.* (2016) have suggested the need to open the
debate on a new future balance between the competing water uses. More recently, based on climate projections
obtained from Coupled Model Intercomparison Project Phase 5 (Taylor *et al*. 2012), Dayon *et al.* (2018) have
shown a significant increase in hydrological drought severity with a meridional gradient (up to -55% in southern
France for both the annual minimum monthly flow with a return period of 5 years and the mean summer river
flow) while a more uniform increase in agricultural drought severity is projected over France for the end of the
21st century.
The challenges associated with possible impact of climate change on droughts have received increasing attention
by researchers, stakeholders and policy makers in the last decades. To date climate change impact studies are
usually dedicated to water resources (e.g., Vidal *et al.* 2016, Collet *et al.* 2018, Hellwig and Stahl 2018, Samaniego
*et al.* 2018) or water needs for the competing users (e.g., Bisselink *et al*. 2018). However, examining the suitability
of regulatory instruments, such as Drought Management Plans, is also essential to establish successful adaptation
strategies. These plans state which type of water restrictions should be imposed to non-priority uses during severe
low-flow events; under climate change, those water restrictions and stakeholders' access to water resources might
need to be revised as drought patterns and severity might change. In most climate change impact studies, analyses
on the regulatory measures are often limited to maintaining environmental flows – especially when assessing future
hydropower potential. To date, no climate change impact on water regulatory measures have yet been assessed at
the regional scale, highlighting a gap in developing robust adaptation plans. This study aims to address this gap by
suggesting a framework, applying it to southeastern France and publishing the associated results.
The paper develops a framework to simulate legally-binding water restrictions (WR) under climate change in
the Rhone-Méditerranée district (southeastern France) and to assess the likelihood of future restrictions depending
on their sensitivity, performance and exposure to climate deviations. The approach is adapted from the risk-based
approaches such as developed in parallel by Brown *et al*. (2011) −named "Decision Tree Framework" −and
Prudhomme *et al*. (2010) −named "Scenario neutral approach"−and aims to establish a ranking of areas vulnerable
to climate change in terms of water access for agricultural uses  . This research is a scientific contribution to the
ongoing decade 2013–2022 entitled "Panta Rhei – Everything Flows" initiated by the International Association of
Hydrological Sciences and more specifically to the "Drought in the Anthropocene" working group
(https://iahs.info/Commissions--W-Groups/Working-Groups/Panta-Rhei/Working-Groups/Drought-in-the-
Anthropocene.do, Van Loon *et al*. 2016). Legally-binding water restrictions and their associated decision-making
processes are important for the blue water footprint assessment at the catchment scale.
The paper is organized in four parts. Sect. 2 introduces the area of interest and the source of data. Sect. 3 is a
synthesis of the mandatory processes for managing drought condition implemented within the Rhône-Méditerranée
district and the related water restriction orders adopted over the period 2005-2016. Sect. 4 describes the general
modelling framework developed to simulate WR decisions. The approach is implemented at both local and
regional scales and results discussed in Sect. 5 before drawing general conclusions in Sect. 6.
**2 Study area and materials**
**2.1 Study area**
The Rhone-Méditerranée district covers all the Mediterranean coastal rivers and the French part of the Rhône
River basin, from the outlet of Lake Geneva to its mouth (Fig. 1). Climate is rather varied with a temperate
influence in the north, a continental influence in the mountainous areas and a Mediterranean climate with dry and
hot summers dominating in the south and along the coast. In the mountainous part (in both the Alps and the
Pyrenees) the snowmelt-fed regimes are observed in contrast to the northern part under oceanic climate influences,
where seasonal variations of evaporation and precipitation drive the monthly runoff pattern (Sauquet *et al.* 2008).
Water is globally abundant but unevenly between the mountainous areas, the northern and southern parts of the
Rhône-Méditerranée (RM) district and water resources are under high pressure due to water abstractions. For the
period 2008-2013, annual total water withdrawal was around 6 billion of $m^3$ in the (excluding any water abstraction
for energy such as cooling nuclear plants and hydropower) with a more than used for irrigation (3.4 billion of $m^3$,
including 2 billion of $m^3$ for channel conveyance). Use for public and industrial supply is of 1.6 and 1 billion of
$m^3$, respectively. Because of an intense competition for water between different users — agricultural, municipal,
and industrial — and the environment, some areas within the RM district can be vulnerable during low-flow
periods. Around 40% of the RM district suffers from water stress and scarcity (http://www.rhone-
mediterranee.eaufrance.fr/gestion/gestion-quanti/problematique.php) and has been identified by the French RM
Water Agency as areas with persistent imbalance between water supply and water demand.
**2.2 Drought management plan**
Drought management plans (DMPs) define specific actions to be undertaken to enhance preparedness and
increase resilience to drought. In France DMPs include regulatory frameworks to be applied in case of drought,
named "arrêtés cadres sécheresse". The past and operating DMPs and the water restriction orders were inspected
in the 28 departments of the RM district. They were obtained from:
-    The database of the DREAL Auvergne-Rhône-Alpes ("Direction Régionale de l'Eau, de l'Alimentation et du

Logement" in French) including WR levels and duration at the catchment scale available over the period 2005-

2016 within the RM district;

-    The online national database PROPLUVIA (http://propluvia.developpement-durable.gouv.fr) with WR levels

and dates of adoption at the catchment scale for the whole France available from 2012.

The most recent consulted documents date from January 2017.
**2.3 Hydrological data**
The hydrological observation dataset is a subset of the 632 French near-natural catchments identified by
Caillouet *et al.* (2017). Daily flow data from 1958 to 2013 were extracted from the French HYDRO database
(http://hydro.eaufrance.fr/). Time series with more than 30% of missing values or more than 30% of null values

were disregarded. Finally, the total dataset consist of 106 gauged catchments located in the RM district with minor human influence and with high quality data. The selected catchments are benchmark catchments where near natural drought events are observed and current water availability is monitored. Water can be abstracted from other nearby streams.

A selection of 15 evaluation catchments (Table 1) were used to calibrate and to evaluate the Water Restriction Level modelling framework (Sect. 4), selected because (*i*) they have complete records of stated water restriction, including dates and levels of restrictions - which was not the case of other catchments, and (*ii*) they are located in areas where water restriction decisions are frequent. To facilitate interpretation, the 15 catchments have been ordered along the north-south gradient. The Ouche and Argens River basins (n°1 and 15 in Table 1) are the northernmost and the southernmost gauged basins, respectively. The 15 catchments encompass a large variety of river flow regimes according to the classification suggested by Sauquet *et al.* (2008) (see Appendix A) that can be observed in the RM district (e.g., the Ouche (1 in Table 1, pluvial regime), Roizonne (3, transition regime) and Argens (15, snowmelt-fed regime) River basins).

**2.4 Climate data**

Baseline climate data were obtained from the French near-surface Safran meteorological reanalysis (Quintana-Seguí *et al.* 2008; Vidal *et al.* 2010) onto an 8-km resolution grid from 1 August 1958 to 2013. Exposure data was based on the regional projections for France (Table 2) available from the DRIAS French portal (www.drias-climat.fr, Lémond *et al*. 2011). Catchment-scale data were computed as weighted mean for temperature and sum for precipitation based on the river network elaborated by Sauquet (2006).

**3 Operating Drought Management Plans in the Rhône-Méditerranée district**

The French Water Act amended on September 24, 1992 (decree n°92/1041) defines the operating procedures for the implementation of drought management plan (DMP). Following the 2003 European heat wave, drought management plans including water restrictions have been gradually implemented in France (MEDDE 2004). Water restrictions fall within the responsibility of the prefecture (one per administrative unit or department), as mentioned in article L211-3 II-1° of the French environmental code. Their role in drought management is to ensure that regulatory approvals for water abstraction continuously meet the balance between water resource availability and water uses including needs for aquatic ecosystems. *De facto*, legally-binding water restrictions have to fulfill three principles: (*i*) being gradually implemented at the catchment scale in regard with low-flow severity observed at

various reference locations, (*ii*) ensuring users equity and upstream-downstream solidarity and (*iii*) being time-
limited to fix cyclical deficits rather than structural deficits. The prefecture is in charge of establishing and
monitoring the DMP operating in the related department.
Past and current drought management plans were analyzed to identify the past and current modalities of
application, the frequency of water restriction orders and the areas affected by water restrictions. Gathering and
studying the regulatory documents was a tedious in particular because of their lack of clear definition of the
hydrological variables used in the decision-making process.
This analysis shows that the implementation of the DMPs has evolved for many departments since 2003, e.g.,
with changes in the terminology and a national scale effort to standardize WR levels. Now severity in low-flows
is classified into four levels, which are related to incentive or legally-binding water restrictions. These measures
affect recreational uses, vehicle washing, lawn watering and domestic, irrigation and industrial uses (Table 3).
Level 0 (named "vigilance") refers to incentive measures, such as awareness campaign to promote low water
consumption from public bodies and general public. Levels 1 to 3 are incrementally legally-binding restriction
levels; level 1 (named "alert") and 2 (named "reinforced alert") enforcing reductions in water abstraction for
agriculture uses, or several days a week of suspension; level 3 (named "crisis") involves a total suspension of water
abstraction for non-priority uses, including abstraction for agricultural uses and home gardening, and authorizes
only water abstraction for drinking water and sanitation services. Due to change in the naming of WR levels since
their creation one task was dedicated to restate the WR decisions (hereafter "OBS") since 2005 with respect to the
current classification into four WR levels.
For all catchments, a WR decision chronology was derived, showing a large spatial variability in WR (Fig. 1) -
note that the 15 evaluation catchments (Table 1) are located in the most affected areas. Between 2005 and 2012,
WR decisions were mainly adopted between April and October (98% of the WR decisions, Fig. 2), with 62% in
July or August, peaking in July.
Decisions for adopting, revoking or upgrading a WR measure are taken after consultation of "drought
committees" bringing the main local stakeholders together, the meeting frequency of which is irregular and
depends on hydrological drought development. The adopted restriction level is mainly based on the existing
hydrological conditions at the time, *i.e.*, based on low-flow monitoring indicators measured at a set of reference
gauging stations and their departure from a set of regulatory thresholds. This varies greatly across the RM district
(Fig. 3). The low-flow monitoring indicators usually considered are:
-    the daily discharge $Qdaily$,
-    the maximum discharge $QCd$, for a window with length $d$ days, $QCd(t)=\max(Qdaily(t'), t' \in [t-d+1, t])$ and
-    the mean discharge $VCd$, for a window with length $d$ days, $VCd(t) = \frac{1}{d} \int_{t-d+1}^{t} Qdaily(t')dt'$ .
Both $QCd$ and $VCd$ are computed over the whole discharge time series on moving time windows with duration
$d$ associated with WR decision varying between 2 and 10 days depending on DMPs. $VC3$ (40% of DMPs) and
$QC7$ (17% of DMPs) are the most commonly used, but other single indicators include $Qdaily$ (17%), $QC5$ (14%),
$QC10$ (8%), $QC2$ (3%), $VC10$ (3%), and with mixed indicators also used (e.g., 14% of $VC3$ and $Qdaily$ together.
The threshold associated with WR also varies within the district, generally associated with statistics derived
from low-flow frequency analysis, but also fixed to locally-defined ecological requirements. In the context of
DMPs, series of minimum $QCd$ or $VCd$ are calculated by the block minima approach and thereafter fitted to a
statistical distribution. The block is not the year but the month or given by the division of the year into 37 10-
day time-window. The regulatory thresholds are given by quantiles with four different recurrence intervals
associated to the four restriction levels. Generally, return periods $T$ of 2, 5, 10 and 20 years are associated with
the "vigilance", "alert", "reinforced alert" and "crisis" restriction levels, respectively. For example, let us
consider thresholds based on the annual monthly minima of $VCNd$. The block minima approach is carried out
on the $N$ years of records for each month $i$, $i=1\ldots,12$ leading to twelves datasets {$min\{VCNd(t), month(t)=i,$
$year(t)=j\}$, $j=1,\ldots,N$}. The twelve fitted distribution allows the calculation of 48 values of thresholds (=12
months $\times$ 4 levels) with four $T$-year recurrence intervals.
The meteorological situation is also examined in terms of precipitation deficit and likelihood of significant
rainfall event considering available short to medium-range weather forecasts. There are heterogeneities in the
drought monitoring variables, the time period on which deficit is calculated and the permissible deviation from
long term average values.
Where appropriate, other supporting local observations such as groundwater levels, reservoir water levels,
field surveys provided by the ONDE network (Beaufort *et al.* 2018) or feedbacks from stakeholders can be used
to inform final decisions.
Since their creation, DMPs have been frequently updated regarding the definition of the regulatory thresholds
and the monitoring variables, the water uses affected by legally-binding restrictions, the selection of the monitoring
sites, etc. It was especially done following the publication of the circular of the French ministry of Ecology in May
2011, and updates often occur after a year with a severe drought to include feedbacks and lessons for the future.
Decision-making processes is definitely heterogeneous in both time and space, which does not make the WR
modelling easy. In addition, official texts stating the DMPs were not all available for this study. Facing this
complexity, simplifying assumptions will be considered in the modelling framework presented in Section 4.3.4
Risk-based framework and the related tools.
**3 Risk-based framework and the related tools**
**4.1 The scenario neutral concept**
Traditionally, hydrological impact studies are often based on "top down" (scenario-driven) approaches, easy to
interpret, but with associated conclusions becoming outdated as new climate projections are produced. In addition
scenario-based studies may fail to match decision-making needs since the implication in terms of water
management is usually ignored (Mastrandrea *et al.* 2010). As a substitute to scenario-driven approach, the
scenario-neutral approach (Brekke *et al.* 2009, Prudhomme *et al.* 2010, 2013a, 2013b, 2015, Brown *et al.* 2012,
Brown and Wilby 2012, Culley *et al.* 2016, Danner *et al.* 2017) has been developed to better address risk-based
decision issues. The suggested framework shifts the focus on the current vulnerability of the system affected by
changes and on critical thresholds above which the system starts to fail to identify possible maladaptation strategies
(Broderick *et al.* 2019). Applied to water management issues, the scenario-neutral studies (Weiß 2011, Wetterhall
*et al.* 2011, Brown *et al.* 2011, Whateley *et al.* 2014) aim at improving the knowledge of the system's vulnerability
to changes and at bridging the gap between scientists and stakeholders facing needs in relevant adaptation strategy.
Prudhomme *et al.* (2010) have suggested combining of the sensitivity framework with 'top-down' projections
through climate response surfaces. This approach has been applied to low-flows in the UK (Prudhomme *et al.*
2015) and its interests have been discussed as a support tool for drought management decisions.
The risk-based framework adopted contains three independent components (Fig. 4):
(i)      Sensitivity analysis (Fronzek *et al.* 2010) based on simulations under a large spectrum of perturbed

climates to (a) quantify how policy-relevant variables respond to changes in different climate factors,

and (b) identify the climate factors to which the system is the most sensitive. Addressing (a) and (b)

may help modelers to check the relevance of their model (e.g., unexpected sensitivity to a climate factor

regarding the know processes influencing the rainfall-runoff transformation). From an operational

viewpoint, it may encourage stakeholders to monitor in priority the variables that affect the system of

interest (reinforcement of the observation network, literature monitoring, etc.),

(ii)       Sustainability or performance assessment, aiming to identify under which climate (or other) conditions

(e.g., no rain period in spring, heat wave in summer, etc.) the system fails. A key-challenge in bottom-

up framework is to define performance metrics and associated critical thresholds relevant for the system

of interest. In the case of our study, these thresholds will make it possible to distinguish duration of

water restrictions, which are unacceptable for users,

(iii)      Exposure, as defined by state-of-the-art regional climate trajectories superimposed to the climate

response surface. The exposure measures the probability of changes occurring for different lead times

based on available regional projections.

All the components of the framework together contribute to the vulnerability of the system (including its
management) to systematic climatic deviations.

The sensitivity analysis was conducted applying a water restriction modelling framework. Climate conditions

were generated applying incremental changes to historical data (precipitation and temperature) and introduced as
inputs in the developed models to derive occurrence and severity of water restriction under modified climates. The
tool chosen here to display the interactions between water restriction and the parameters that reflect the climate
changes is a two-dimensional response surface, with axes represented by the main climate drivers. This
representation is commonly used in scenario neutral approach. For example, in both Culley *et al*. (2016) and Brown
*et al*. (2012) the two axes were defined by the changes in annual precipitation and temperature. When changes
affect numerous attributes of the climate inputs, additional analyses (e.g., elasticity concept combined with
regression analysis (Prudhomme *et al*. 2015), Spearman rank correlation and Sobol' sensitivity analyses (Guo *et*
*al*. 2017)) may be required to point out the key variables with the largest influence on water restriction that form
thereafter the most appropriate axes for the response surfaces.

Performance assessment is a challenging task for hydrologists since it requires information on the impact of

extreme hydrometeorological past events on stakeholders' activities. Simonovic (2010) used observed past events
selected with local authorities on a case study in southwestern Ontario (Canada), chosen for their past impact
(flood peak associated with a top-up of the embankments of the main urban center; level II drought conditions of
the low water response plan). Schlef *et al.* (2018) set the threshold to the worst modelled event under current
conditions. Whateley *et al.* (2014) assessed the robustness of a water supply system and the threshold is fixed to
the cumulative cost penalties due to water shortage evaluated under the current conditions. Brown *et al.* (2012)
and Ghile *et al.* (2014) suggested selecting thresholds according to expert-judgment of unsatisfactory performance
of the system by stakeholders, whilst Ray and Brown (2015) use results from benefit-cost analyses. The spatial
coverage of a large area, such as the RM district, and the heterogeneity in water use (domestic needs, hydropower,
recreation, irrigation, etc.) makes it challenging for a systematic, consistent and comparable stakeholder
consultation to be conducted and for a relevant critical threshold $T_c$ to be fixed for all the users. Facing this
complexity, only the irrigation water use will been examined here, since it is the sector which consumes most
water at the regional scale, with a critical threshold defined for this single water use.
Exposure to changes here is measured using regional projections, visualized graphically by positioning the
regional projections in the coordinate system of the climate response surfaces and identifying the associated
likelihood of failure relative to $T_c$. Note that, to update the risk assessment, only the exposure component has to
be examined (including the latest climate projections available onto the response surfaces).
**4.2 The rainfall-runoff modelling**
The conceptual lumped rainfall-runoff model GR6J was adopted for simulating daily discharge at 106 selected
catchments of the RM district. The GR6J model is a modified version of GR4J originally developed by Perrin *et*
*al.* (2003), well suited to simulate low-flow conditions (Pushpalatha *et al.* 2011). The 4-parameter version of the
model GR4J has been progressively modified. Lemoine (2008) has suggested a new groundwater exchange
function and a new routing store representing long-term memory in the GR5J model. Pushpalatha *et al.* (2011)
finally introduced in the GR6J model an exponential store in parallel to the existing store of the GR5J model.
Considering additional routing stores is consistent regarding the natural complexity of hydrological processes, and
in particular, the dynamics of flow components in low flows (Jakeman *et al.*, 1990).
The GR6J model has six parameters to be fitted (Fig. 5): the capacity of soil moisture reservoir (X1) and of the
routing reservoir (X3), the time base of a unit hydrograph (X4), two parameters of the groundwater exchange
function F (X2 and X5) and a coefficient for emptying exponential store (X6). The GR6J model is combined here
with the CemaNeige semi-distributed snowmelt runoff component (Valéry *et al.* 2014). The catchment is divided
into five altitudinal bands of equal area on which snowmelt and snow accumulation processes are represented. For
each band, daily meteorological inputs – including solid fractions of precipitation - are extrapolated using elevation
as covariate and the snow routine is calculated separately. Finally, its outputs are then aggregated at the catchment
scale to feed GR6J. The two parameters of CemaNeige S1 and S2 control the snowpack inertia and the snowmelt,
respectively. S1 is used to compute the thermal state of the snow pack $eTG$, which is an equivalent to the internal
snowpack temperature (°C). $eTG(t)$ at day $t$ is a weighted linear combination of the value of $eTG(t-1)$ ($\times$S1) and
the air temperature at the day $t$ ($\times$(1-S1)). S2 is the snowmelt degree-day factor used to calculate the daily snowmelt
depth by multiplying the air temperature when it exceeds 0°C, with S2. The splitting coefficient of effective rainfall
between the two stores (SC, in Fig. 5) has been fixed to 0.4 by Pushpalatha *et al.* (2011) since calibrating SC lead
to only slight better performance. The allocation of the outflow from the soil moisture reservoir in 90% as
percolation and 10% as surface and sub-surface runoff in the GR6J model is the results of previous studies. The
GR6J model was selected for its good performance across a large spectrum of river flow regimes (e.g., Hublart *et*
*al.* 2016, Poncelet *et al.* 2017).
No routine to simulate water management (e.g., reservoir) was considered here since discharges of the 106
gauging stations are weakly altered by human actions or naturalized discharges (*i.e.* flows corrected from the
effects of water use). The eight parameters (six from the GR6J model and two from the CemaNeige module) were
calibrated against the observed discharges using the baseline Safran reanalysis as input data and the Kling–Gupta
efficiency criterion (Gupta *et al.* 2009) $KGE_{SQRT}$ calculated on the square root of the daily discharges as objective
function. The $KGE_{SQRT}$ criterion was used to give less emphasis of extreme flows (both low and high flows). As
the climate sensitivity space includes unprecedented climate conditions (including colder climate conditions
around the current-day condition), the CemaNeige module was run for all the 106 catchments even for those not
currently influenced by snow.
The two step procedure suggested by Caillouet *et al.* (2017) was adopted for the calibration: first the eight free
parameters were fitted only for the catchments significantly influenced by snowmelt processes – *i.e.*, when the
proportion of snowfall to total precipitation > 10% - and second, for the other catchments, the medians of the
CemaNeige parameters were fixed and the six remaining parameters are then calibrated. Calibration is carried out
over the period 1 January 1973 to 30 September 2006 with a 3-year spin-up period to limit the influence of reservoir
initialization on the calibration results. The criterion $KGE_{SQRT}$ and the Nash-Sutcliffe efficiency criterion on the
log transformed discharge $NSE_{LOG}$ (Nash and Sutcliffe 1970) were calculated over the whole period 1958-2013
for the subset of 15 evaluation catchments (Table 1), showing $KGE_{SQRT}$ and $NSE_{LOG}$ values are above 0.80 and
0.70 respectively. These two goodness-of-fit statistics indicate that GR6J adequately reproduces observed river
flow regime, from low to high flow conditions. The less satisfactory performances of GR6J are observed for the
Tarn and Roizonne River basins, both characterized by smallest drainage areas and highest elevations of the
dataset. These lowest performances are likely to be linked to their location in mountainous areas (snowmelt
processes are difficult to reproduce) and to their size (the grid resolution of the baseline climatology fails to capture
the climate variability in the headwaters).
**4.3 The water restriction level modelling framework**
The Water Restriction Level (WRL) modelling framework developed aims to identify periods when the
hydrological monitoring indicator is consistent with legally-binding water restrictions. Only physical components
(mainly hydrological drought severity) leading to WR decisions are considered, with no socio-political factor
accounted for to model water restrictions.
To enable comparison of results across all catchments – in particular to combine response surfaces obtained
from different catchments (see Section 5.1) - the same drought monitoring indicators and regulatory thresholds
were adopted in all the catchments (see Section 3 for details), selected as most commonly used in the 28 DMPs
across the RM district, specifically $VC3$ as monitoring indicator and $10d\text{-}VCN3$ with return periods $T$ of 2, 5, 10
and 20 years as regulatory thresholds. Each regulatory threshold is defined for a 10-day calendar period between
1[st] April and 31[st] October, resulting in 21 sets of four thresholds. Water restrictions are decided after consulting
drought committees that convene irregularly depending on hydrological conditions over a time window, *i.e.*, the
last $N$ days. Here a time window for analysis of $N= 10$ days was decided, which is consistent with the prefectural
decision-making time frame (frequency of updates in water restriction statements). The WRL modelling time-step
is finally fixed to 10 days and a representative value of WRL is given to the 21 10-day calendar periods from April
to October. Thus WRL is thus computed as follows:
-   $VC3(t)$ is computed from daily discharge $Qdaily(t)$ every day $t$;
-   $VC3(t)$ is compared to the corresponding regulatory thresholds to create time series of daily water

restriction level *wrl*, with $wrl(t)$ ranging from 0 ('no alert') to 3 ('crisis'):

o   if $10d\text{-}VCN3(2) \geq VC3(t) > 10d\text{-}VCN3(5)$, $wrl(t)=0$

o   if $10d\text{-}VCN3(5) \geq VC3(t) > 10d\text{-}VCN3(10)$, $wrl(t)=1$

o   if $10d\text{-}VCN3(10) \geq VC3(t) > 10d\text{-}VCN3(20)$, $wrl(t)=2$

o   if $10d\text{-}VCN3(20) \geq VC3(t)$, $wrl(t)=3$

-  A *WRL(d)* time series is created as the median of *wrl(t)* for each 10-day period;
-  The *WRL(d)* value is set to zero if preceding 10-day precipitation total exceeds 70% of inter-annual

precipitation average( precipitation correction).

Inputs of the WRL model are daily discharges and precipitation. Outputs are WRL time series with values for each
21 10-day calendar period from April to October. Modelling is only applied to the period April-to-October, the
irrigation period and when most water restrictions are put in place. The low-flow monitoring indicator *VC*3 and
the regulatory thresholds $10d\text{-}VCN3(T)$ are computed from daily discharge time series *Qdaily* based on full period
of records prior to 31$^{st}$ December 2013. The log-normal distribution is used to assess the return periods.
The WRL modelling framework can be applied to both observed and simulated time series. For the later, outputs
from GR6J are used for simulations under current and modified climate conditions. Regulatory thresholds are
derived from simulated discharge using the Safran baseline meteorological reanalysis as input, to moderate the
possible effect of bias in rainfall-runoff modelling.
The WRL modelling framework was verified in the 15 evaluation catchments (Table 1). WRL simulations based
on modelled (hereafter "GR6J") and observed (hereafter 'HYDRO') discharge were compared graphically to
official WR measures ("OBS"). A further assessment was conducted using the *Sensitivity* and *Specificity* scores
(Jolliffe and Stephenson 2003) to examine how well the WRL modelling framework can discriminate WR severity
levels (Table 4). The *Sensitivity* score assesses the probability of event detection; the *Specificity* score calculates
the proportion of "No" events that are correctly identified. An event was defined as any legally-binding Water
Restriction of at least level 1, and 'non-event' a period where WRL is 0 or without WR. Comparisons were made
over the 2005-2013 period, corresponding to the common period of availability for OBS, HYDRO and GR6J.
Fig. 6 shows years with severe simulated WRLs (e.g., 2005 and 2011) and years with no or few simulated WRs
(e.g., 2010 and 2013). Both GR6J and HYDRO simulations are generally consistent with OBS, even if misses are
found (e.g., basins 9 to 11 during the year 2005). There is no systematic bias, with some overestimations (e.g.,
2005 using GR6J in basins 1 and 15; 2007 using HYDRO in basin 15), underestimations (e.g., 2009 in basin 6, 7,
and 8) and misses (e.g., 2005 using HYDRO in basin 1).
*Sensitivity* and *Specificity* scores computed with OBS considered as benchmark (Fig. 7) show a large variation
across the catchments, in particular for *Sensitivity*. *Specificity* scores are around 0.85 for both GR6J and HYDRO,
suggesting that more than 85% of the observed non-events were correctly simulated by the WRL modelling
framework. The median of WRL *Sensitivity* score with HYDRO is around 45%, indicating that for half the
catchments, less than 45% of observed events are detected based on HYDRO discharges, but this raises to 68% of
events detected when WRLs are simulated based on GR6J discharge. Using GR6J is more effective for detecting
legally-binding restriction than using observed discharges while it is less efficient for predicting periods without
restriction for most of the catchments. There is a compensatory effect, which is not easy to detect graphically since
*Sensitivity* scores are more sensitive than *Specificity* scores due to the reduced number of observed days with
adopted restrictions. No evidence of systematic bias associated with catchment location or river flow regime was
found: northern (blue) and southern (red) catchments are uniformly distributed in the *Sensitivity*/*Specificity* space.
*Sensitivity* and *Specificity* scores using HYDRO as benchmark in the contingency table were also used to
compare simulations from GR6J discharge with those obtained from HYDRO discharge. Median values reach
84% (*Sensitivity*) and 92% (*Specificity*), showing high consistency between HYDRO and GR6J. No statistical link
between hydrological model and WRL model performance was found, with $R^2$ between $NSE_{LOG}$ and *Sensitivity*,
or $NSE_{LOG}$ and *Specificity* lower than 7%. In addition, the similar skill scores of GR6J and HYDRO modelling
suggest that possible biases in rainfall-runoff modelling does not impact on the ability of the WRL modelling
framework to correctly simulate declared or not declared WRs.
Choosing the same definitions for the monitoring indicator and regulatory thresholds is a simplifying assumption
and may partly explain the deviations between simulated (HYDRO or GR6J) and adopted (HYDRO) WR
measures. Before stating for *VC*3 and 10*d-VCN*3 the four prevalent modalities found in the current DMPs have
been tested to reproduce observed WR and results has shown a weak sensitivity to the hydrological variables
considered in the WR modelling framework. The mains reasons are that all the indicators and thresholds are
derived from *Qdaily* time series, are highly correlated and thus share, above all, the same information on the
dynamics and on the severity of drought.
Heterogeneity in basin characteristics and rules imposed by the DMPs should not result in a systematic difference
in *Sensitivity* and *Specificity* score between GR6J and HYDRO identified for most of the 15 evaluation catchments.
Simulations were made on near pristine catchments and thus water uses are unlikely to be the main reason. Other
causes of higher *Sensitivity* scores obtained when simulated discharges are used as input have been investigated in
the WRL modeling framework. However, results of this analysis have not been conclusive. The aforementioned
tests with the four prevalent modalities have all led to higher *Sensitivity* score using GR6J and higher *Specificity*
score using HYDRO, demonstrating that the choice of the monitoring indicator and regulatory thresholds is
probably not involved. A "smoothing" introduced by the hydrological modelling was also suspected but
autocorrelation in observed and GR6J simulated $VC3$ time series was found very similar. Future works may re-
investigate these aspects. They will need to explore new ones (e.g., the way $WRL$ is derived from the daily values
$wrl$ for each 10-day period) using a longer verification period with not necessary uniform but fixed regulatory
framework. Indeed some catchments have experienced only three years with legally-binding water restrictions and
DMP have been frequently during the 2005-2013 period (see the black vertical segments in Fig. 6).
Discrepancy between simulated and adopted WR measures is most likely due to the other factors involved in the
making-decision process. When regulatory thresholds are crossed, restrictive measures should follow the DMPs.
In reality, the measures are not automatically imposed, but are the result of a negotiating process. This process
includes for example some expert-judgment factors such as (*i*) the evolution of low-flow monitoring indicators
and thresholds over the years (e.g., annual revision for the Ouche, and irregular revision for the Isère (38), Gard
(30), Alpes-de-Haute-Provence (04) and Lozère (48) departments (last one in 2012)); (*ii*) the role of drought
committees in negotiating a delay in WR level applications to limit economic damages or to harmonize responses
across different administrative sectors sharing the same water intake; (*iii)* the local expertise especially regarding
the uncertainty in flow measurements (Barbier *et al*. 2007) impacting on the low-flow monitoring indicators, e.g.,
Cote d'Or (21) and Lozère (48) in the northern and southwestern parts of the RM district, respectively. Note that
where WR decisions are not uniquely based on hydrological indicators but also involve a negotiation process, the
results of the WRL modelling framework should be interpreted as potential hydrological conditions for stating
water restrictions.
Results of our sample study on 15 evaluation catchments show deviations for most catchments, but links between
order restrictions and hydrological drought severity. These deviations may partly be attributed to the use of the
same monitoring indicator and regulatory thresholds across the catchments in the modelling (whilst it is not true
in reality), as a necessary assumption for a region scale analysis. Tests with $QC7$ as low-flow monitoring variable
combined with the two dominant modalities for the regulatory thresholds show a weak sensitivity of the WRL
modelling skill to the choice of the indicators (with a slight increase in *Specificity* score (~ 90%) while *Sensitivity*
score is reduced (< 50%) using GR6J). Whilst the developed WRL modelling framework does not account for
expert-decision brought by drought committees - and hence is not designed to simulate the exact WR decisions -
its ability to simulate 68% of the stated restrictions over the period 2005-2013 demonstrates its usefulness as a tool
to objectively simulate the potential of drought restrictions based on hydrological drought physical processes. The
methodology was applied to the 106 catchments of the RM district under climate perturbations to assess the
potential impact of climate change on water restriction in the region. The resulting analysis focuses on water
restriction level higher than 1, denoted thereafter WR*.
**4.4 The generation of perturbed climate conditions**
The generation of climate response surfaces relies on synthetic climate time series representative of each explore
climate condition, and used as input to the impact modelling chain (here hydrological model and WRL modelling
framework). Methods based on stochastic weather simulation have been used (Steinschneider and Brown 2013,
Cipriani *et al*. 2014, Guo *et al*. 2016, 2017), but they can be complex to apply in a region with such heterogeneous
climate as the RM district. Alternatively, the simple "delta-change" method (Arnell 2003) has been commonly
used to provide a set of perturbed climates in scenario-neutral approach (Paton *et al*. 2013, Singh *et al*. 2014), and
was used here, similarly to (Prudhomme *et al*. 2010, 2013a, 2013b, 2015).
Following Prudhomme *et al.* (2015), monthly correction factors $\Delta P$ and $\Delta T$ are calculated using single-phase
harmonic functions:
$$\Delta P(i) \; = \; P_0 \; + \; A_P \cdot \cos\left[(i - \varphi_P) \cdot \frac{\pi}{6}\right]. \tag{1}$$

$$\Delta T(i) \; = \; T_0 \; + \; A_T \cdot \cos\left[(i - \varphi_T) \cdot \frac{\pi}{6}\right]. \tag{2}$$

with $P_0$ and $T_{0+}A_T$ mean annual changes in precipitation (1) and temperature (2), respectively; $i$ indicator of the
month (from 1 to 12); $\varphi_P$ the phase parameter and $A_p$ the semi-amplitude of change (e.g., half the difference
between highest and lowest values). These corrections factors were applied to the baseline climate data sets to
create perturbed daily forcings:
$$P^*(d) = \; P(d) \cdot [\overline{PM}(\text{month}(d)) + \Delta P(\text{month}(d)]/\overline{PM}(\text{month}(d)) \tag{3}$$

$$T^*(d) = \; T(d) + \; \Delta T\big(\text{month}(d)\big) \tag{4}$$

with $P(d)$ and $T(d)$ baseline precipitation and temperature values for day $d$; $P^*(d)$ and $T^*(d)$ the corrected (or
perturbed) values for day $d$; $\overline{PM}(\text{month}(d))$ average monthly baseline precipitation for month($d$). Corrected
potential evapotranspiration $PET^*$time series were derived from temperature values using the formula suggested
by Oudin *et al* (2005):
$$\text{PET*}(d) = \max\left[\text{PET}(d) + \frac{Ra}{28.5}\frac{\Delta T(\text{month}(d))}{100}; 0\right] \qquad\qquad (5)$$
with $\text{PET}(d)$ baseline potential evapotranspiration values for day $d$; $Ra$ extra-terrestrial global radiation for the
catchment.
The baseline climate (precipitation and temperature) time series were extracted from the Safran reanalysis over
the period 1958-2013 (56 years), and perturbed time series generated for the same length. The range of climate
change factors to generate the perturbed series were chosen to encompass both the range and the seasonality of
RCM-based changes on projections in France. A set of 45 precipitation and 30 temperature scenarios was created
(Fig. 8), spanning the range of potential future climate suggested by Terray and Boé (2013) and combined
independently, resulting in a total of 1350 precipitation and temperature perturbations pairs used to define the
climate sensitivity space. In this application,
-    $P_0$, (mm.an$^{-1}$)= -20 +20/3× ($j$-1) , $j$= 1,.., 9 ,
-    $Ap$ (mm.season$^{-1}$) = 20/3× ($j$-1) , $j$= 1,.., 5 ,
-    $T_0$ (°C.an$^{-1}$) = $j$-1 , $j$= 1,…, 6 ,
-    $A_T$ (°C.season$^{-1}$) = -0.5 +2×($j$-1) , $j$= 1,…,5 ,
-    $\varphi_P$   parameter is fixed to 1 to consider minimum change in January and maximum change in July and
-    $\varphi_T$  is fixed to 2 to get maximum change in August.

**4.5 The assumptions on water uses**

Water uses and the feedbacks between use and available resources are not explicitly addressed in this application,
either under current or future conditions. This should not be considered as a limitation for basins where
hydrological modelling has been implemented. Indeed, the 106 basins under study have been carefully chosen
since they are currently little or not influenced by human actions. These catchments are benchmark catchments
where natural water availability is monitored for the statement of restriction orders. Water can be abstracted from
other neighboring rivers. Water needs will probably evolve in the next decades. Water requirement for irrigation
may increase in parallel to air temperature or may decrease due to adaptive actions (e.g. farmers may choose to
plant specific crops less sensitive to water shortages). Water needs and sensitivity to water restrictions depend on
socio-economic and institutional pathways. Forward-looking studies have been recently carried out with the
involvement of local experts but at the local scale (Grouillet *et al.* (2015) for the Hérault River basin; Andrews
and Sauquet (2016) for the Durance River basin). The distinct underlying assumptions make difficult to combine
and to extend the prospective scenarios over the RM district. Thus, the water restriction modelling framework
considers, in this application, the "Business-as-usual" scenario, which assumes that only minor change in water
demand behavior will occurs. In particular, no major alteration of the river flow regime is projected for the 106
catchments. Despite unrealistic, maintaining the current conditions allows assessing the impact of climate change
regardless of any other human-induced changes. The advantage is that results are easier to understand and to
embrace by stakeholders than those obtained with complex multi-sectorial scenarios they may not identify with.
**5 Drought management plans under climate change and their impact on irrigation use**
**5.1 The Water Restriction response surfaces**
The 1350 sets of perturbed precipitation, temperature and PET time series were each fed into the WRL modelling
framework for each 106 catchments. Both $VC3$ (monitoring indicators) and $10d\text{-}VCN3(T)$ (regulatory thresholds)
were computed from GR6J 56 years discharge simulations. For each scenario, the number of 10-day periods under
Water Restriction of at least level 1 (WR*) were calculated, and expressed as deviation from the simulated baseline
value: ΔWR*, hence removing the effect of any systematic bias from the WRL modelling framework. Results are
shown as WR response surfaces built with $x$- and $y$-axes representing key climate drivers. Because different climate
perturbation combinations share the same values of the key climate drivers, hence represented at the same location
of the response surface, the median ΔWR* from all relevant combinations is displayed as color gradient, with the
standard deviation $Sd$ of ΔWR* showed as size of the symbol.
Response surfaces based on different climate variables for $x$ (precipitation) and $y$ (temperature) were generated
over full or part of the water restriction period (April to October "AMJJASO", March to June "MAMJ"; and July
to October "JASO", the latter coinciding with the highest temperatures) and visually inspected to identify the
greatest signal pattern, combined with the smallest dispersion around the surface response (*i.e.*, analysis of the
median and the maximum of $Sd$ values over the grid cells).
The response surfaces are exemplified on three of the 15 evaluation catchments (Table 1, Fig. 9):
-    The Argens River basin, along the Mediterranean coast, severe low-flows occur in summer and actual

evapotranspiration is limited by water availability in the soil,

-    The Ouche River basin, in the northern part of the RM district, has a typical pluvial river flow regime under

oceanic climate influences, where runoff generation is less bounded by evapotranspiration processes,

-     The Roizonne River basin, in the Alps, typical of summer flow regime controlled by snowmelt, with spring

to summer climate conditions dominating changes in low-flows.

The visual inspection of response surfaces shows that:
-     $\Delta$WR* are differently driven by the changes in precipitation $\Delta P$ and in temperature $\Delta T$: $\Delta$WR* is very

sensitive to $\Delta P$ in the Argens River basin (horizontal stratification in the response surface) and to $\Delta T$ in the

Roizonne River basin (vertical stratification in the response surface) whilst being controlled by both drivers

in the Ouche River basin;

-     There is a high likelihood of increase in the duration of water restriction in the Roizonne River basin, as

showed a response surface dominated by positive $\Delta$WR*;

-     *Sd* values may vary significantly from one graph to another (Table 5). For both the Argens and Roizonne

River basins, largest *Sd* are found when the response surfaces are displayed with climate variables computed

over the whole period April-to-October (AMJJASO) while smallest *Sd* are associated with $\Delta$P and $\Delta$T

drivers from March to June. Changes in mean spring to early summer precipitation and temperature mainly

govern changes in WR* for these two basins. Conversely changes in precipitation $\Delta$P and temperature $\Delta$T

over the full period April-to-October seem the dominant drivers of changes in WR* for the Ouche River

basin.

**5.2 Response surface analysis at the regional scale**
Following (Köplin *et al*. 2012, Prudhomme *et al*. 2013a), the 106 response surfaces were classified to define
typical response surfaces, designed as tools to help prioritizing actions for adapting water management rules to
future climate conditions in the RM district. Here a hierarchical clustering based on Ward's minimum variance
method and Euclidian distance as similarity criteria (Ward 1963) was applied and four classes were identified after
inspection of the agglomeration schedule and silhouette plots (Rousseeuw 1987). A manual reclassification was
conducted for the few catchments with negative individual silhouette coefficients to ensure higher intra-class
homogeneity. For each class, a mean response surface and associated *Sd* was computed, and main climate drivers
associated with WR changes identified (Table 5).
All suggest an increase in the occurrence of legally-binding water restrictions when precipitation decreases or
when temperature increases (Fig. 10). Additional temperature increase and its associated *PET* increase can
compensate for precipitation increase and lead to decrease in ΔWR* with intra-class differences emerging in the
magnitude of changes. The identified four typical Water Restriction response surfaces show a weak regional
pattern and common features. Class 4 (including the Roizonne River basin) regroups snowmelt-fed river flow
regimes in the Alps, whilst basins of Class 1 are mainly Mediterranean river flow regimes. Class 2 (including the
Ouche River basin) and Class 3 catchments are partly influenced by both precipitation and temperature, with
ΔWR* in Class 2 catchments less sensitive to climatic changes (flatter WR response surface) than catchments of
Class 3. Flow regime of Classes 2 to 3 ranges from rainfall-fed regimes with high flow in winter and low flow in
summer in the northern part of the RM district to regimes partly influenced by snowmelt with high-flows in spring
in the Alps and in the Cevennes.
To further the regional analysis and help sensitivity assessment at un-modelled catchments, basin descriptors
were investigated as possible discriminators of the four classes. A set of potential discriminators - which included
measures of the severity, frequency, duration, timing and rate of change in low-flow events (Table 6), the drainage
area and the median elevation for the catchment and one climate descriptor (mean annual precipitation and mean
annual potential evapotranspiration used to compute an aridity index) – were introduced in a CART model
(Classification And Regression Trees, Breiman *et al.* 1984), aimed at performing successive binary splits of a
given data set according to decision variables. Through a set of "*if-then*" logical conditions the algorithm
automatically identifies the best possible predictors of group membership, starting from the most discriminating
decision variable to the less important factors. The optimal choices are fixed recursively by increasing the
homogeneity within the two resulting clusters. At each step one of the clusters (node) is divided into two non-
overlapping parts. Here, to free results from catchment size influence, descriptors related to severity were
expressed in mm/year, mm/month or mm/day.
Results show three top discriminators, the aridity index being the strongest:
- Aridity index *AI* given by the mean annual precipitation divided by the mean annual potential
evapotranspiration (UNEP, 1993),
- Baseflow index *BFI*, a measure of the proportion of the baseflow component to the total river flow, calculated
by the separation algorithm separation suggested by Lyne and Hollick (1979),
- Concavity Index *IC* (Sauquet and Catalogne 2011) to characterize the contrast between low-flow and high-
flow regimes derived from quantiles of the flow duration curve,
CART overall misclassification (18%) suggests a satisfactory performance in classification method,
characterized by a parsimonious algorithm (five nodes and three variables) with potential for a first guess
assessment of the WR response to disruptions and evaluation of the robustness of existing water restriction at the
department-level scale. For each class, Fig. 11 shows the empirical distribution of the three main discriminators,
the mean timing $\theta$ of daily discharge below $Q95$ and its dispersion $r$, based on circular statistics, where $Q95$ is the
95[th] quantile derived from the flow duration curve.
The classification discriminates catchments primarily on the seasonality of low-flow conditions and the aridity
index, with the extreme classes (1 and 4) being particularly well discriminated.
Geographically, Class 1 catchments are mainly located along the Mediterranean coast and include the Argens
River basin; ∆WR* is mainly driven by changes in precipitation in spring and early summer. Class 1 gathers water-
limited basins with small values of $AI$ and a weak sensitivity to climate change in summer. In these dry water-
limited basins, the mid-year period exhibits the minimal ratio $P/PET$ and changes in summer precipitation has
hence only a moderate impact on low-flows; spring is the only season when $PET$ changes are likely to result in
both actual evapotranspiration and discharge changes. WR levels are more likely controlled by antecedent soil
moisture conditions in spring and early summer. This behavior is typical of the basins under Mediterranean
conditions and was discussed in the context of a scenario-neutral study in Australia (Guo *et al*. 2016). For those
catchments, climate drivers computed in spring (over the period MAMJ) are used to describe the x- and y-axes of
the response surface, fully consistent with water-limited basin processes.
Catchments of both Class 2 and 3 have similar $IC$, hence suggesting that flow variability is not a proxy for low-
flow response to climatic deviation. However, $BFI$ values for Class 3 are lower than for Class 2 while Class 3 is
characterized by high values for $AI$. Despite higher capability to sustain low-flows (see $BFI$ values) the response
surface representative of Class 2 is more contrasted than that of Class 3; a possible reason could be drier conditions
under current conditions (the median of $AI$ equals 2.5 for Class 3 against 1.6 for Class 2). The monthly perturbation
factors (see Sect. 5.1) are the same for all the classes but the changes in relative terms are less significant regarding
the current climate conditions for Class 3 than for Class 2, and may explain the limited changes in river flow
patterns.
Class 4 regroups catchments with low flows in winter and significant snow storage. The $BFI$ values are high and
due to smooth flow duration curves, $IC$ demonstrates also high values.

**5.3 Risk assessment at the basin scale**

The risk-based framework has been applied to the irrigation water use since annual net total water withdrawal for agriculture purposes is ranked first at the regional scale. Note that in the Rhône-Méditerranée district around 90% and 10% of water used for irrigation originate from surface water and groundwater, respectively. To complement water needs irrigators may also have access to small reservoirs (storage capacity usually less than 1 Mm$^3$). Most of the reservoirs are filled by surface water in winter and release water later in the following summer. Water restrictions are not imposed to these reservoirs but it is assumed here that during severe drought events the majority of them are empty and thus the existence of potential sources auxiliary to surface water on the conclusions has limited influence on the conclusions.

We assumed here that irrigated farming is globally under failure if the duration with limited or suspended abstraction is above a critical threshold $T_c$ that causes insufficient water for crops. The catchment or area $i$ will be considered more vulnerable than the catchment or area $j$ if the likelihood of failure (*i.e.*, exceeding $T_c$) for catchment or area $i$ is more than the likelihood of failure for catchment or area $j$. The critical threshold $T_c$ is a value of total number of days with legally-binding water restrictions that needs to be fixed. To move closer to reality and following Simonovic (2010), the value of $T_c$ is based on the analysis of past events. A possible way to fix $T_c$ is to simulate historic drought events observed during the period 2005-2012 and the effects of water restrictions on crop yield and quality and on economic losses. Computing water deficits was considered rather tricky at the farming scale - partly due to the high heterogeneity in crop and soil types, watering systems, conveyance efficiencies, etc. across the RM district - and we have investigated the use of 'agricultural disaster' notifications as proxies to identify the damaging conditions instead.

Specifically the 'agricultural disaster' notifications are issued by the agriculture ministry following recommendations from the Prefecture to each department affected by extreme hydro-meteorological events, and applied uniformly over the RM district. Whilst 'agricultural disaster' status is a global index that may mask heterogeneity in crop losses within each department, and that reflects losses related to both agricultural and hydrological droughts, it has the advantage of being directly related to economic impact, and uniformly applied across the RM district, hence suitable for a regional-scale analysis. The national system of compensation to farmers is initiated for areas notified under 'agricultural disaster' status.

Over 2005-2012, only one agriculture disaster was declared, in 2011, and applied to 70 of the 95 departments in
continental France, and to 16 of the 28 departments fully or partly located in the RM district. Data are collected
by the French Ministry of Agriculture and Food and they are not publically available. The year 2011 was the only
year when the national system of compensation has been triggered between 1958 and 2013 and the analysis of
simulated water restrictions for this year fixed the value for $T_c$. The duration of water restrictions was calculated
individually for each catchment and converted into anomalies $\Delta WR^*(2011)$ with respect to the benchmark value
(mean over the period 1958-2013). For consistency with the indicators used in the response surfaces, this threshold
$\Delta WR^*(2011)$ is derived from GR6J outputs.
The RCM-based projections of all the catchments of the class for the three time slices 2021-2050, 2041-2070
and 2071-2100 were superimposed to the representative response surfaces to assess the risk of failure (Fig. 4).
Finally the vulnerability resulting from the combination of the three components sensitivity, performance and
exposure was measured by the proportion of RCM-based projections leading to critical situations, similarly to
Prudhomme *et al*. (2015). Technically this Vulnerability Index (*VI*) calculated as the proportion of exposure
simulations that fail below the critical threshold $T_c$ is the complement to the "climate-informed" robustness index
(*CRI*) (Whateley *et al*. 2014). Given one specific climate projection, a catchment or a group of catchments could
be judged vulnerable if on average $T_c$ is exceeded. *VI* is introduced here to account for the uncertainty in climate
projections in risk assessment. This index should be interpreted as conditional probability (risk) with respect to a
specified ensemble of future climates.
Fig. 12 shows an application to the Ouche River basin, North of the RM district (1, Fig. 1, Table 1) and declared
under agricultural disaster status in 2011. The black dotted line are isopleths connecting points of the response
surface with $\Delta WR^* = \Delta WR^*(2011) = T_c$ (= 7 10-day periods for this catchment), and delimits the climate space
leading to median climatic situations more severe than 2011 ($\Delta WR^* > \Delta WR^*(2011)$, above left) or less severe than
2011 ($\Delta WR^* < \Delta WR^*(2011)$, below right) $\Delta WR^*(2011)$. As reference, the black solid line ($\Delta WR^* = 0$) delimits
the climate space associated with more (above left) or less (bottom right) water restrictions compared with the
whole period average (1958-2013). Basin-scale exposure projections (Table 2) were plotted onto the WR response
surface for three time-slices 2021-2050, 2041-2070 and 2071-2100 (grey symbols), showing a warmer trend but
no total precipitation signal. Whilst by the end of the century, projections move towards the critical threshold
$\Delta WR^*(2011)$ climate space, pointing out a significant increase in more severe low-flows, there remain a large
spread in signal (dispersion of the grey symbols) and the vulnerability index equals zero for this catchment.

**5.4 A regional perspective for prioritizing adaptation strategies**

Following the methodology applied to the Ouche River basin, $\Delta WR^*(2011)$ were calculated for individual catchments and averaged to produce a value of $T_c$ relevant for each Class (Table 7). Class variation in $\Delta WR^*(2011)$ is large, with Class 2 and 3 showing thresholds of at least 7 10-day periods, whilst they are close to zero for Class 1 and Class 4. The scatter in the $\Delta WR^*(2011)$ values is certainly due to heterogeneity in crops, in irrigation systems, in climate conditions, etc. at the regional scale leading to locally differentiated sensitivity to water restrictions as well as to biases in WR modelling. Since only the year 2011 it is now difficult to conclude on the origins of the dispersion (natural or non-natural). However the distribution and absolute values of the critical thresholds reflect well the spatial pattern of WR enforced from May to September 2011, with Southern regions and the French Alps moderately affected by lack of rainfall in spring compared to the Northern and Western regions of the RM district (Fig. 13). Surprisingly negative values for $\Delta WR^*(2011)$ are found for some catchments of Classes 1 and 4, providing no evidence to support their agricultural disaster status that year. At the RM scale, average $\Delta WR^*(2011)$ equals 38 days when considering all catchments, and increases to 66 days when considering only catchments under agricultural disaster status. Simplifying but realistic assumptions are imposed by the lack of detail information; thus only one value was considered at the regional scale despite high dispersion in $\Delta WR^*(2011)$ values (Table 7): the critical threshold $T_c$ was set to the average of the $\Delta WR^*(2011)$ values computed on all catchments in departments under agricultural disaster status in 2011 (6.6 10-day periods), and was used thereafter for all classes. Note that this value of $T_c$ seems realistic: it represents a significant period with restrictions (66 days or 30% of the time between the 1$^{st}$ April and the 31$^{st}$ October).

Using the Class WR response surface as diagnostic tools, exposure information (grey symbols) and thresholds ($\Delta WR^*=0$, solid, $\Delta WR^*(2011)$, dashed black lines) were displayed (Fig. 14), and *VI* calculated (Table 7). The location of the two isopleths $\Delta WR^* = \Delta WR^*(2011)$ (black dotted line) and $\Delta WR^* = 0$ (black straight line) in the WR response surface depends on the shape of the response surface and differ from one class to another. The portion of the WR response surface associated with $\Delta WR^*<0$ is gradually lower from Class 1 to Class 4 suggesting that catchments of Class 4 are more subject to an increase in water restriction occurrence than catchments of the other classes. Classes 1 and 4, the most extreme responses classes, contain fewer catchments, whilst Classes 2 and 3, characterized by an intermediate response, have the most of the catchments. Because of the large geographical spread of catchments of Class 2 and 3, an expert-based division was done to distinguish catchments with continental (northern sectors) and Mediterranean (southern sectors) climate in terms of exposure. This is to better

capture the predominantly north–south gradient in future projections of both temperature and rainfall, as they
differing impact on the river flow regime (e.g., Boé *et al*. 2009; Chauveau *et al*. 2013; Dayon *et al*. 2018). For all
classes, vulnerability increases with lead time, with Class 4 showing the largest vulnerability and Class 1 being
the less vulnerable despite its location in the Mediterranean area. In the two classes 2 and 3, vulnerability increases
from North to South in the RM district ($VI$ = 13% for Class 2-N against 32.9% for Class 2-S at the end of the
century). These contrasted results are mainly explained by the difference between exposure characterizations since
a common value of the threshold $T_c$ was adopted.
**5.4 Water restriction policy implementation**
In 2011, France adopted a general framework for action—the French National Climate Change Impact
Adaptation Plan ("Plan National d'Adaptation au Changement Climatique (PNACC)" in French)—with numerous
recommendations related to research and observation. Five priorities of the first PNACC related to water resources
have been highlighted. The PNACC has been recently reviewed and the PNACC2 published in December 2018
confirms the place of DMPs as tools for monitoring water resources and water allocation, and for driving greater
public and stakeholder awareness (https://www.ecologique-solidaire.gouv.fr/adaptation-france-au-changement-
climatique).
However and until now, impacts of future climate change is not account for in DMPs. The development of DMPs
have helped to ease past conflicts at the department scale. Water users are now facing more frequent water
restrictions (more than half France have departments experiencing WR ≥ 1 between 2011 and 2018 (Fig. 15))  and
the timing and the level of the restrictions vary from one year to another: the highest number of French departments
with WR ≥ 1 was observed in summer in both 2015 and 2017 while the year 2018 was characterized by late water
restrictions (mostly in autumn). Stakeholders are now questioning the DMP implementation, but only at the short
term – the impact of climate change is not yet a subject matter. One of their main concerns is the heterogeneity in
current restrictions levels and timing from one department to another or from the upstream to the downstream part
of the catchment. One of the option being considered to address this challenge in southeastern France is to
harmonise the definition of the regulatory thresholds, at the regional scale. Results obtained here show that the
standardisation will probably not fix the problem due to the balance between socio-political and hydrological
factors in the final WR statement.
The map displaying the class membership could be a convenient tool for local authorities to discuss the spatial
heterogeneity in terms of impact to drought on water restrictions under both current and future climate conditions.
Despite operating rules uniformly applied, there is a high variability in catchments responses within the department
(see the southernmost department in Fig. 10). Therefore, any investigation on DMPs at the department level
disregarding this heterogeneity will be biased. The sensitivity analysis provides information for local authorities
to better understand the differences in catchment responses to observed droughts in areas, which fall within their
responsibility. For instance, water management in basins of Class 4 could be more problematic during a year with
a severe heat wave while it could be more problematic for a year with a pronounced precipitation deficit for
catchments of Class 1. It is likely that the differences in the impact of droughts on WR will persist if stakeholders
do not question the assumption of a uniform definition for the hydrological indicators within the department.
DMPs have been recognized in the PNACC as relevant water management tools and our findings have also
implications for adaptation strategies. We have shown that the climate change effects could be felt more acutely
during the irrigation period by an increase in water restriction. Thus, relying on surface water to compensate
deficits is highly hazardous. Options under consideration are saving water, enhancing water storage by building
new small dams or securing water access by transferring water from the Rhone River (e.g., Ruf 2012), which is
considered as an "overabundant" river within the RM district. Saving water is the solution favoured by the RM
Water Agency. Creating new storages is increasingly considered as potential solution to secure water for
agriculture since they are not subject to water restrictions. Authorising new water storages may also reduce the
sense of unfairness among users in areas with no secured access. Most of the small reservoirs are filled by surface
water in winter, release water later in summer for irrigation purposes and then limit the pressure on water resource
during crises. However, there is actually a wide discussion about these hydraulic structures in France since their
cumulative impacts on the ecosystem and their efficiency are not well known (Habets *et al.* 2018). Building
adaptation strategies on additional water storage may lead to maladaptation since natural inflows will probably
decrease, and delay the mutation of agricultural practices and conservation measures. In addition, there is actually
no guarantee that these reservoirs will be filled and that their storage capacity will be enough to cope with severe
droughts.
The RM Water Agency has taken other the objectives of PNACC at the regional scale and has initiated an
unprecedented major initiative that provides guidance for the River Basin Management Plan (2016–2021). The
adaptation strategy partly relies on an analysis of the vulnerability in different water-related sectors (water
resources, soil-moisture, biodiversity, and water quality) within the RM district to climate change. The study
complements this former analysis by focusing here on agricultural uses and meets the requirements for
vulnerability assessment carried out by the RM Water Agency: it covers the same area and the methodology is
uniformly applied across the area of interest. It may help the RM Water Agency identifying when and where
actions and investments are the most needed to mitigate the effects of climate change (probably in catchments of
Class 4 from the short perspective, and later for the other areas).
**6 Conclusions**
This paper presents a first attempt to analyse and simulate water restrictions over a large area in France applying
an alternative approach to the classical "top-down" approach. The risk-based approach developed here relies on
sensitivity-based analyses to a wide range of climate changes, making it scenario-neutral. However ex ante climate
projections are introduced in the last stage of the framework to assess the likelihood of failure.
The analysis of the past and current DMPs in the RM district shows a decision-making processes highly
heterogeneous in terms of both low-flow monitoring variable and regulatory thresholds. In reality, the WR
statements follow a set of rules defined in the DMPs (which can be simulated and reproduced automatically) but
also expert judgment or lobbying from key stakeholders - which are not accounted for in the WRL modelling
framework put in place here. However, the post-processing of GR6J outputs allows detecting more than 68% of
severe alerts (more severe than level 1), making the developed framework a useful tool. Our study is a first step
towards a comprehensive accounting of physical processes, but does not capture socio-economic factors, also
critically important and reaches out to interdisciplinary for completing the modelling framework designed here.
The study at the regional scale illustrates an expected difficulty to simulate accurately a regulatory framework.
Further improvement is not expected in enhancing hydrological models but in reproducing decision-making
processes. The overall performance could be improved by scrutinizing the minutes of the drought committees to
better understand the weight of the stakeholders in the final statement.
The sensitivity analysis and the related response surfaces suggest that basins located in the Southern Alps are
the most responsive basins to climate change and that those experiencing a high ratio *P/PET* are found the less
responsive. The classification method CART has been applied to 106 responses surfaces associated with 106
gauged basins and leads to four classes with different sensitivity. The key-variables known at un-modelled but
gauged catchments can be introduced in the decision-tree to finally predict the assignment as a first guess to one
of the four classes. Water managers are thus encouraged to monitor in priority and more accurately temperature
and/or precipitation when and where the sensitivity of their catchments is found the highest. This may mean efforts
to reinforce field instrumentation within these key catchments.
Although incomplete, the proposed framework demonstrates, as expected (see Assessment Box SPM.2 Table 1
in (IPCC 2014)), a sensitivity of the DMPs to climate changes. The impact of climate change on the river flow is
expected to be gradual, thus offering opportunities to update, to harmonize and to adapt Drought Management
Plans to changes in climate conditions and water needs. As a consequence, the need for adaptation of existing
drought action plans could differ much from one catchment to another and should take into account intrinsic
sensitivity to climate change besides 'top-down' projections. Results also show needs to firstly adapt DMPs in
temperature sensitive catchments more subject to a significant increase in legally-binding restrictions in the short
term. In contrast, the capacity to anticipate changes in both the occurrence and severity of WR, and their
consequences for water management will be challenging in catchments where water restrictions are mainly driven
by precipitation due to their high uncertainties in future regional climate projections.
The risk-based approach was applied to assess the vulnerability of irrigation due to regulatory instruments under
modified climate. Evaluating the impact of climate change on irrigation was not the objective of the suggested
framework; it has been applied to estimate the likelihood of failure for irrigation at various lead times, instead.
Usually, a failure can be stated when irrigation water needs are not fully satisfied. This case study suggests the use
of a proxy obtained from a national system of compensation to define a critical threshold (maximum acceptable
duration with water restriction). Analysis, however, was based on limited data (one year) and a better failure
assessment is required using other years (e.g., 2015 and 2017). The higher the probability, the more vulnerable the
irrigation use within the department. Finally, socio-economic system stressors like agricultural practices,
population growth, water demand, etc. should be considered to highlight combinations that would lead to
unacceptable conditions and to assess the performance of various adaptation strategies under an extended set of
future climate conditions (Poff *et al*. 2016).
Climate response surface appears as a convenient tool for simulating and discussing future perspectives locally
on the basin scale or more broadly on a given management territory. For example, they can support implement
adaptive strategies (see - as an example - the Robust Decision Making framework suggested by Lempert and
Groves (2010)): response surfaces can be drawn for different adaptation scenarios combined with periodic updates
of DMPs including rules for defining regulatory thresholds and monitoring variables evolving over time, etc.
Note that all results are based on a single hydrological model, but a multi-model approach could be applied as
the magnitude of the rainfall-runoff response was shown vary with different hydrological models (e.g., Vidal *et*
*al.* 2016; Kay *et al.* 2014). Finally, an extension of the area of interest to the whole France may bring to light a
more complete typology of response surfaces and a wider range of sensitivity.
**Acknowledgments**

The authors thank Météo-France for providing access to the Safran database. Regional projections were obtained

from the DRIAS portal (http://drias-climat.fr/) and consulted on November 2016. Analyses were performed in R
(R Core Team 2016) with packages airGR (Coron *et al.* 2017), chron (James and Hornik 2017), circular (Lund *et*
*al.* 2017), doParallel (Calaway *et al.* 2017), dplyr (Wickham and François 2015), ggplot2 (Wickham 2009),
hydroTSM (Zambrano-Bigiarini 2014), RColorBrewer (Neuwirth 2014), reshape2 (Wickham 2007), rpart
(Therneau *et al.* 2018), scales (Wickham 2016), stringr (Wickham 2017) and zoo (Zeileis and Grothendieck 2005).
The study was funded by Irstea and the French RM Water Agency.

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

| N° | River basin | Department (department number) | Station number | Elevation (m.a.s.l.) | Area (km$^2$) | Regime class | NSE$_{LOG}$ | KGE$_{SQRT}$ |
|---|---|---|---|---|---|---|---|---|
| 1 | Ouche | Côte d'Or (21) | U1324010 | 243 | 651 | 6 | 0.84 | 0.94 |
| 2 | Bourbre | Isère (38) | V1774010 | 202 | 703 | 1 | 0.85 | 0.92 |
| 3 | Roizonne | Isère (38) | W2335210 | 936 | 71.6 | 11 | 0.71 | 0.84 |
| 4 | Bonne | Isère (38) | W2314010 | 770 | 143 | 12 | 0.80 | 0.91 |
| 5 | Buëch | Hautes-Alpes (05) | X1034020 | 662 | 723 | 9 | 0.84 | 0.93 |
| 6 | Drôme | Drôme (26) | V4214010 | 530 | 194 | 3 | 0.81 | 0.89 |
| 7 | | | V4264010 | 263 | 1150 | 9 | 0.85 | 0.88 |
| 8 | Roubion | Drôme(26) | V4414010 | 264 | 186 | 9 | 0.83 | 0.93 |
| 9 | Lot | Lozère (48) | O7041510 | 663 | 465 | 3 | 0.88 | 0.94 |
| 10 | Tarn | Lozère (48) | O3011010 | 905 | 67 | 8 | 0.73 | 0.90 |
| 11 | | | O3031010 | 565 | 189 | 9 | 0.81 | 0.91 |
| 12 | Hérault | Hérault (34) | Y2102010 | 126 | 912 | 8 | 0.83 | 0.88 |
| 13 | Asse | Alpes de Haute-Provence (04) | X1424010 | 605 | 375 | 9 | 0.80 | 0.86 |
| 14 | Caramy | Var (83) | Y5105010 | 172 | 215 | 2 | 0.85 | 0.94 |
| 15 | Argens | Var (83) | Y5032010 | 175 | 485 | 2 | 0.80 | 0.92 |

**Table 1: Main characteristics of the 15 catchments used for validation of water restriction simulations. Station number refers to the catchment number in the HYDRO database and regime class to the classification suggested by Sauquet *et al*. (2008) with a gradient from Class 1- pluvial fed regime moderately contrasted to Class 12- snowmelt fed regime.**


| Data source | Representative Concentration Pathway | | | Reference |
|---|---|---|---|---|
| | RCP2.6 | RCP4.5 | RCP8.5 | |
| ALADIN | A | A | NA | Bubnová et al. (1995). Radnoti (1995) |
| First quartile, median and last quartile of the ensemble EURO-CORDEX results | NA | A | A | Jacob et al. (2014) |
| WRF | NA | A | NA | Skamarock et al. (2008) |

**Table 2: Regional climate projections available in the DRIAS portal (A: available; NA: not available).**


| Level | Name | Recreational | Vehicle washing | Lawn watering | Swimming-pool filling | Urban washing | Irrigation | Industry | Drinking water and sanitation |
|---|---|---|---|---|---|---|---|---|---|
| 0 | Vigilance | × | × | × | × | × | | | |
| 1 | Alert | × | × | × | × | × | × | × | |
| 2 | Reinforced alert | × | × | × | × | × | × | × | |
| 3 | Crisis | × | × | × | × | × | × | × | × |

**Table 3: Uses affected by water restriction according to the drought severity**


| WR* event | | WR level ≥ 1 (Benchmark) | |
|---|---|---|---|
| | | *Yes* | *No* |
| WR level ≥ 1 (Prediction) | *Yes* | hits | false alarms |
| | *No* | misses | correct negatives |

**Table 4: Contingency table for legally-binding restriction (WR*).**


| | *Sd* | Period | | |
|---|---|---|---|---|
| | | AMJJASO | JASO | MAMJ |
| Argens River basin (Class 1) | median | 1.59 | 1.65 | **0.19** |
| | max | 3.32 | 3.69 | **1.21** |
| Ouche River basin (Class 2) | median | **0.63** | 0.78 | 1.10 |
| | max | **1.03** | 1.52 | 1.99 |
| Roizonne River basin (Class 4) | median | 1.12 | 1.32 | **0.64** |
| | max | 1.98 | 2.49 | **0.91** |
| All | median | **0.69** | 0.80 | 0.70 |
| | max | 1.45 | 1.70 | **1.24** |
| Class 1 | median | 1.16 | 1.24 | **0.25** |
| | max | 2.70 | 2.96 | **1.17** |
| Class 2 | median | **0.72** | 0.85 | 0.89 |
| | max | 1.45 | 1.81 | **1.43** |
| Class 3 | median | **0.41** | 0.49 | 0.64 |
| | max | **0.88** | 0.97 | 1.06 |
| Class 4 | median | 0.91 | 1.14 | **0.81** |
| | max | 1.78 | 2.15 | **1.28** |

**Table 5: Summary statistics for standard deviation *Sd* of the grid for different axes.**

| Component of the river flow regime | Hydrological indices |
|---|---|
| Severity | Flow exceeded 95% of the time ($Q95$)<br>Annual minimum 10-day daily mean low flow with a 5-year recurrence interval<br>Annual maximum deficit below threshold Q95 exceeded 20% of time |
| Duration | Annual maximum maximal duration of the continuous sequence of zero flow within the year, exceeded on average every five years ($D80$). Maximum duration of consecutive zero flows ($D$) are sampled by block maxima approach and $D80$ is defined as the empirical 80th percentile of cumulative distribution function of $D$<br>Seasonal recession time scales ($DT$ and $Drec$). This duration is based on the hydrograph defined by the 1-day and 30-day moving average of the 365 long term mean daily discharges, $d = 1,\ldots, 365$ ($Qd$ and $Q30d$, respectively). $Drec$ is defined by the time lapse between the median $Qd50$ and the 90th quantile $Qd90$ of $Qd$ on the falling limb of the hydrograph defined by $Q30d$ and $DT = \ln(Qd50/Qd90)/Drec$ |
| Rate of Change | Ratio $Q95/Q50$<br>Concavity index derived from flow duration curve $(Q10 - Q99)/(Q1 - Q99)$ (Sauquet and Catalogne 2011). This descriptor is a dimensionless measure of the contrast between low-flow and high-flow regimes derived from quantiles of the flow duration curve<br>Baseflow index ($BFI$). $BFI$ is a measure of the proportion of the baseflow component to the total river flow, calculated by the separation algorithm separation suggested by Lyne and Hollick (1979)<br>Class of river flow regime based on average monthly runoff pattern defined by Sauquet *et al.* (2008) (between 1 and 12)<br>Seasonality ratio ($SR$) $SR = Q95_{AMJJASON}/Q95_{DJFM}$ ($SR > 1$ for mountainous catchment) with $Q95_{AMJJASON}$ and $Q95_{DJFM}$ computed on seasonal flow duration curves |
| Frequency | Proportion of years with at least one value below $Q95$ |
| Timing | Mean day of first occurrence of flow below $Q95$<br>Mean and dispersion of the occurrence of flows below $Q95$ within the year ($\theta$ and $r$, $rsin(\theta)$ and $rcos(\theta)$. These two variables are circular statistics. Each day $i$ with zero flow is converted into an angular ($t_i$) and represented by a unit vector with rectangular coordinates ($cos(t_i)$; $sin(t_i)$). The mean of the cosines and sines defines a representative vector. The value for $\theta$ is obtained by calculating the inverse tangent of the angle of the mean vector and the norm of the mean vector provides a measure of the regularity in the dates (a value close to one indicates a high concentration around $\theta$ while a value close to zero indicates no seasonality) |

**Table 6: Hydrological metrics considered to investigate similarity in CART.**

| Class | | Number of catchments (with agricultural disaster status) | Mean ΔWR*(2011) (with agricultural disaster status) (× 10 days) | Vulnerability index VI (%) | | |
|---|---|---|---|---|---|---|
| | | | | 2021-2050 | 2041-2070 | 2071-2100 |
| **1** | **All** | **15 (2)** | **-1.2 (-2.3)** | **6.1** | **11.5** | **6.7** |
| **2** | **All** | **44 (22)** | **5.0 (7.1)** | **6.4** | **11.8** | **21.6** |
| | N | 25 (18) | 6.1 (6.2) | 0 | 0 | 13 |
| | S | 19 (4) | 3.4 (11.3) | 14.8 | 27.3 | 32.9 |
| **3** | **All** | **38 (13)** | **5.4 (8.7)** | **1.7** | **4.5** | **7.9** |
| | N-E | 25 (4) | 3.7 (3.8) | 0.4 | 0 | 4.5 |
| | S-W | 13 (9) | 8.5 (10.8) | 4.19 | 13.3 | 14.4 |
| **4** | **All** | **9 (3)** | **0 (-0.7)** | **18.2** | **45.4** | **47.2** |
| **All** | | **106 (40)** | **3.8 (6.6)** | **5.8** | **12** | **16.7** |

**Table 7: Summary statistics for the mean anomaly ΔWR*(2011) and for the measure of vulnerability *VI* estimated at the regional scale.**

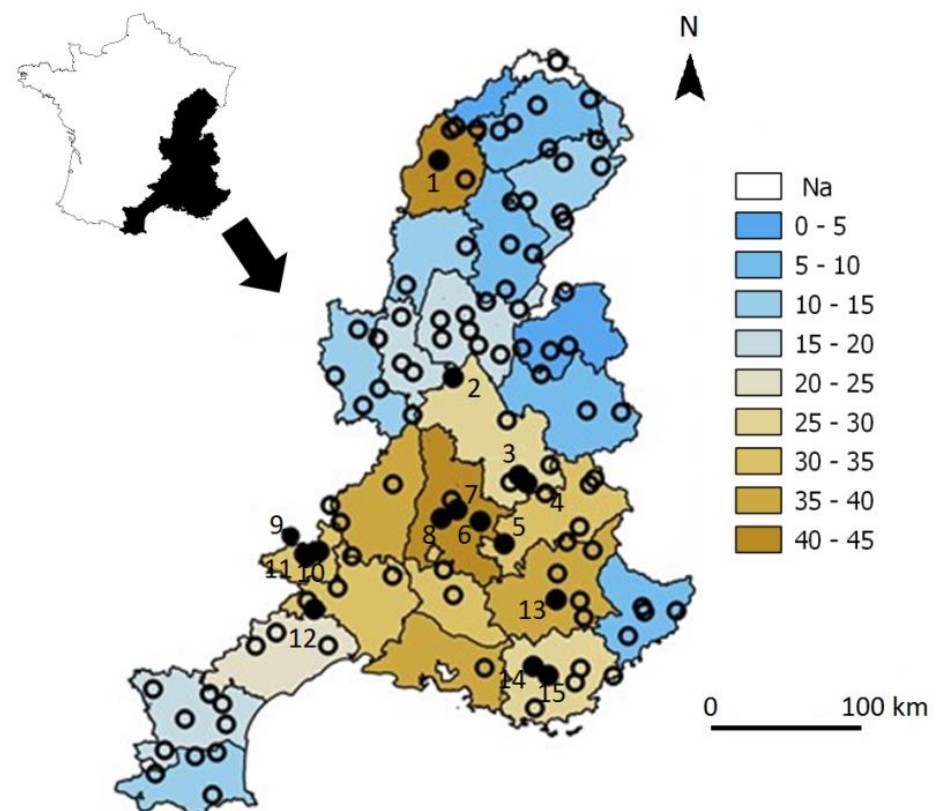

**Figure 1: The Rhône-Méditerranée water district, the total number of WR decisions stated by department over the period 2005-2016 and the gauged catchments ○ where WR decisions are simulated (● denotes the subset of the 15 catchments used for evaluation purposes and the figures are the related ranks presented in Table 1).**

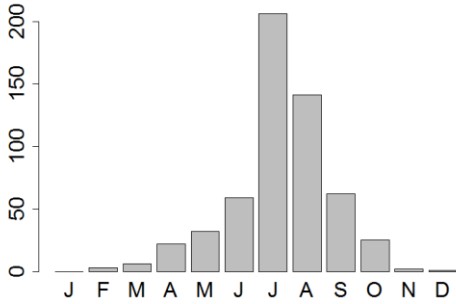


**Figure 2: Total number of stated WR decisions over the RM district per month over the period 2005-2016.**

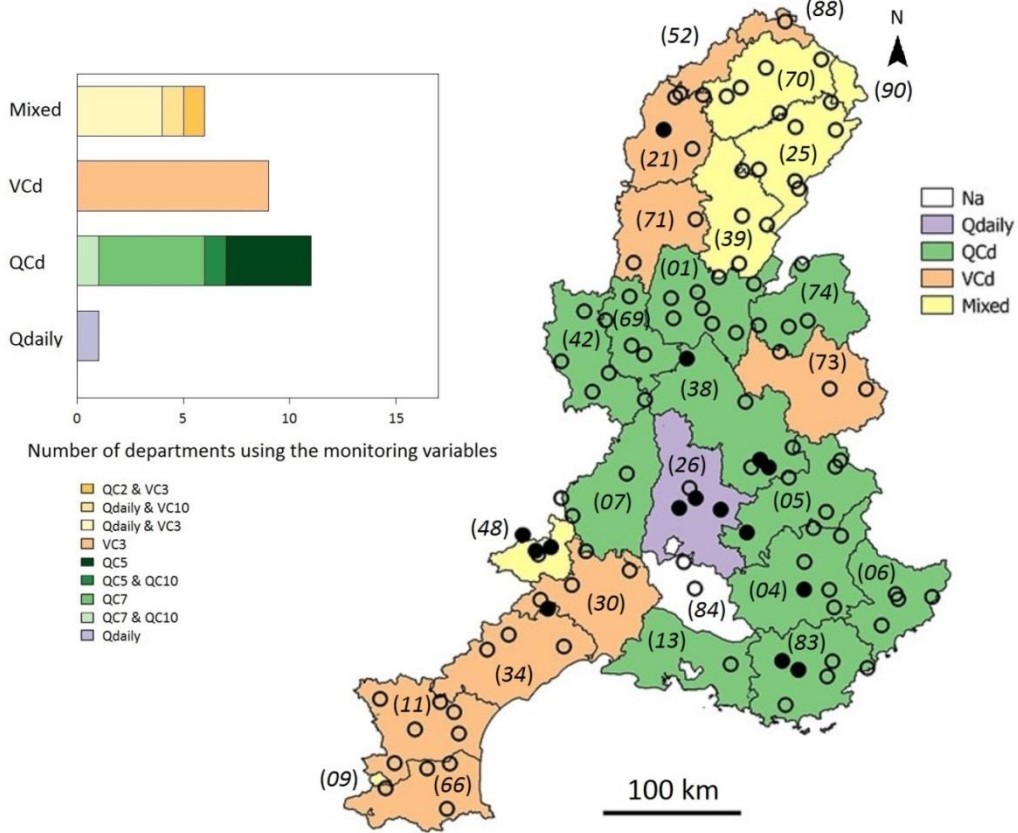


**Figure 3: Low-flow monitoring variables used in the current drought management plans.** *Qdaily* **denotes daily**
**streamflow,** *QCd* **the** *d***-day maximum discharge;** *VCd* **the** *d***-day mean discharge and** *Mixed* **refers to combinations of**
**the aforementioned variables. Department codes are given into brackets.**

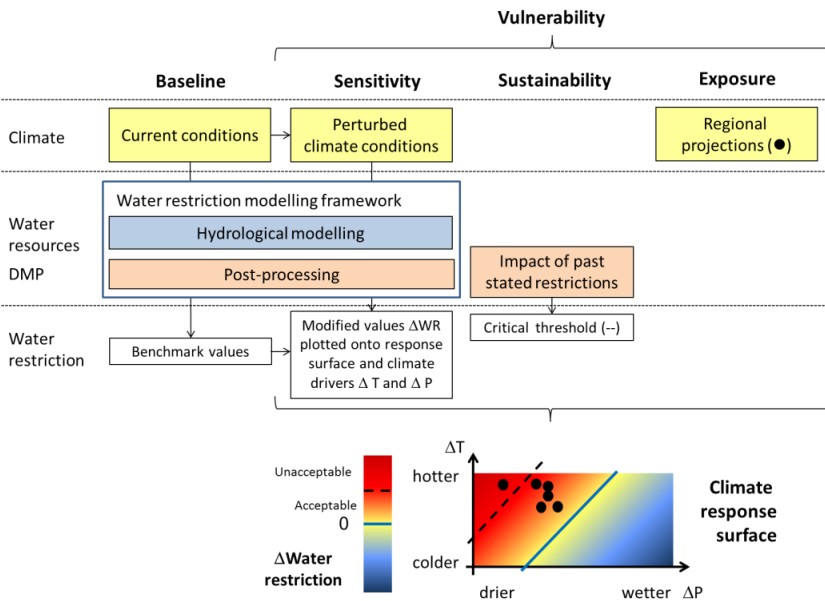


**Figure 4: Schematic framework of the developed approach to assess the vulnerability of the DMPs under climate**
**change.**

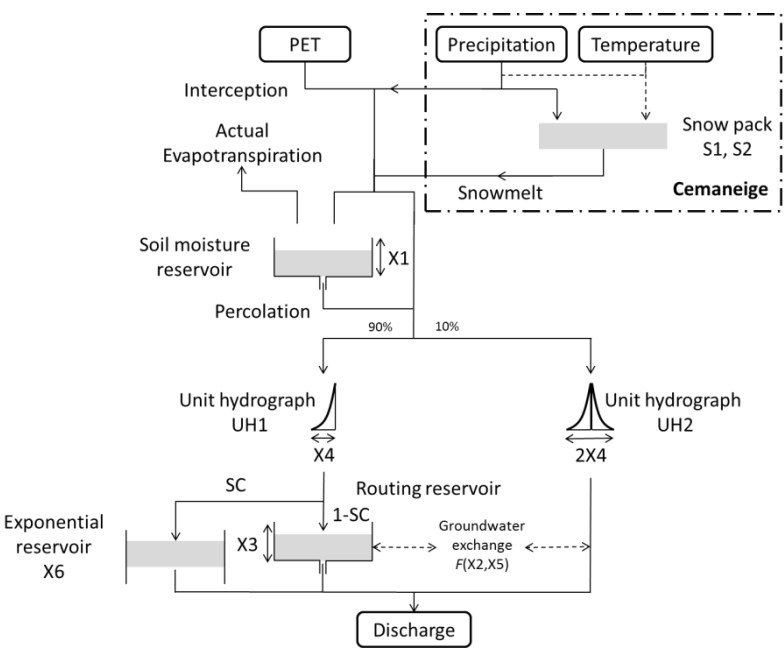


**Figure 5: Schematic of the rainfall-runoff Model GR6J combined with the CemaNeige snowmelt runoff component**
**(after Pushpalatha et al. 2011).**

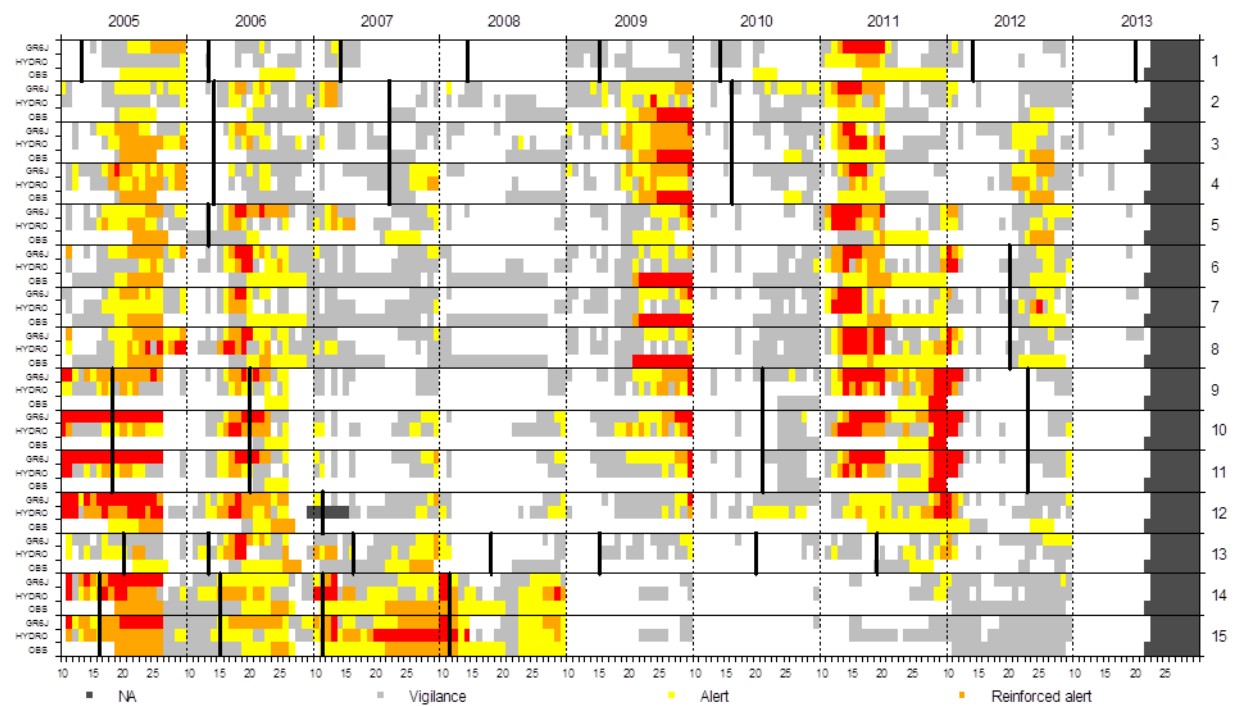


**Figure 6: Observed and simulated water restriction levels considering the two sources of discharge data GR6J and**
**HYDRO for each of the 15 evaluation catchments (Table 1). The x-abscissa is divided into ten-day periods for each year**
**spanning the period April-to-October. Black segments identify updated DMPs.**

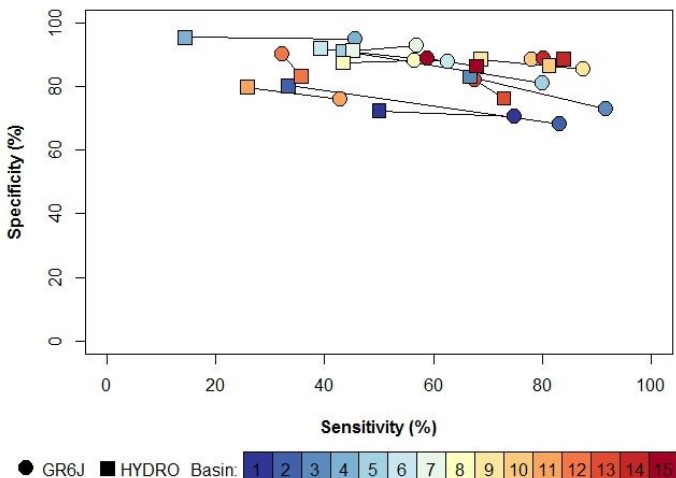


**Figure 7: Skill scores obtained for the WR level model over the period 2005-2013. Each segment is related to one of the**
**15 catchments listed in Table 2. The endpoints refer to the source of discharge data (GR6J or HYDRO).**

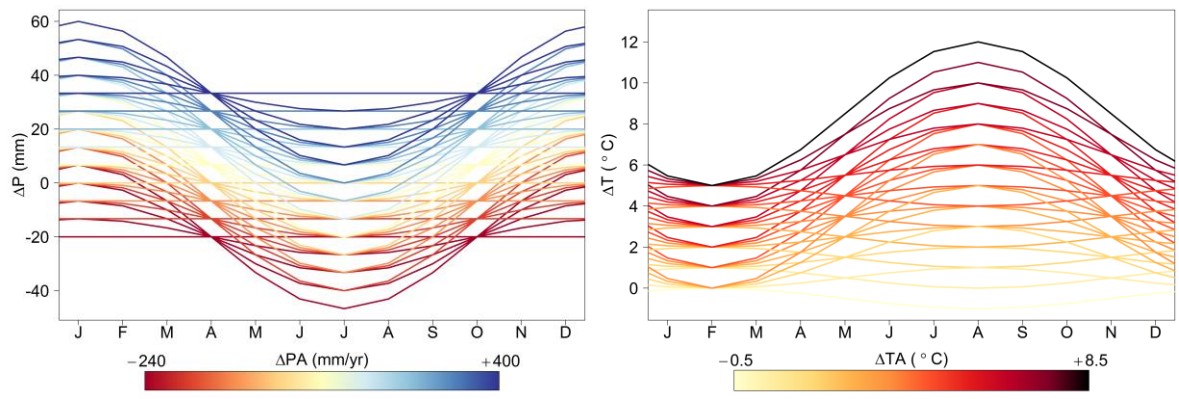


**Figure 8: Monthly perturbation factors ΔP and ΔT associated with the climate sensitivity domain. The color of the line**
**is related to the intensity of the annual change ΔPA and ΔTA.**

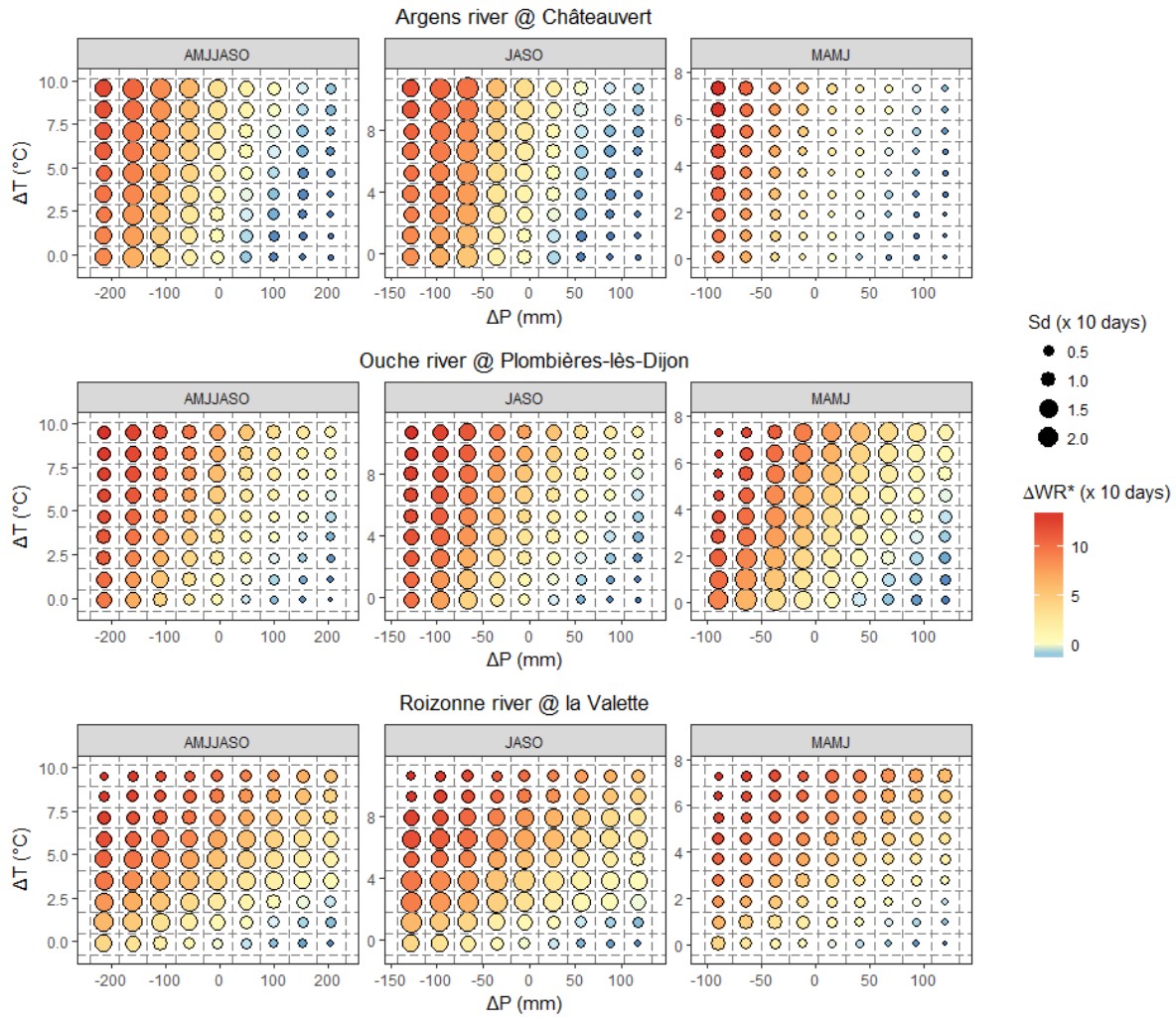


**Figure 9: Climate response surface of legally-binding water restrictions level anomalies ΔWR\* for the Argens, Ouche**
**and Roizonne River basins. Each graph is obtained considering changes in mean precipitation ΔP and temperature ΔT**
**over a specific period as x- and y-axis.**

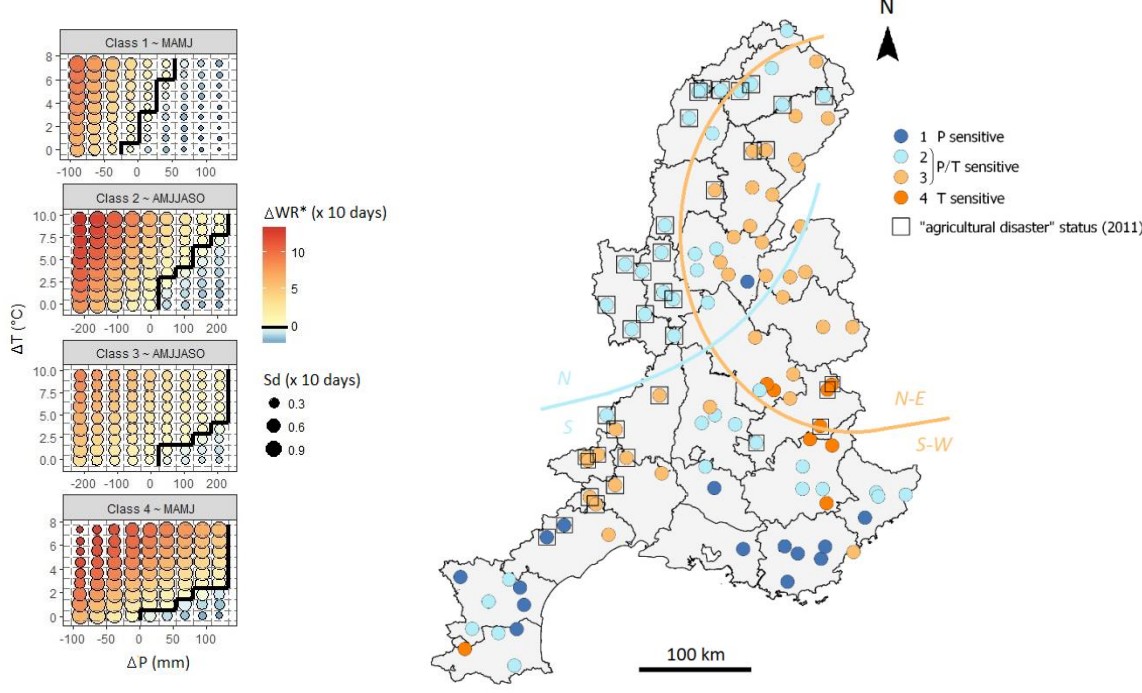


**Figure 10: Results of the hierarchical cluster analysis applied to the climate response surface WR\* level anomalies**

**ΔWR\***



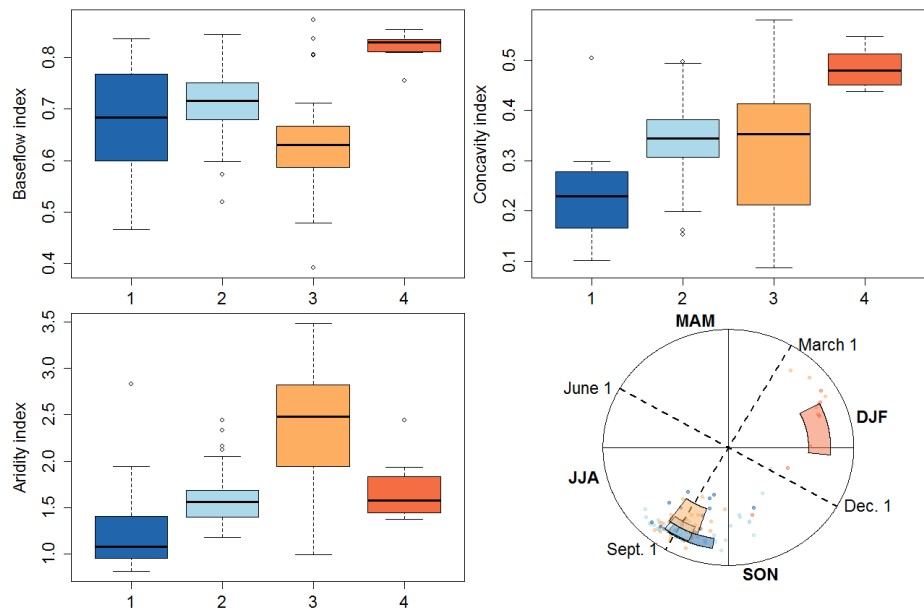


**Figure 11: Statistical distribution of the discriminating factors identified by the CART algorithm (top level, top left and bottom left) and the mean timing $\theta$ of daily discharge below $Q95$ and its dispersion $r$ (bottom right). The boxplots are defined by the first quartile, the median and the third quartile. The whiskers extend to 1.5 of the interquartile range; open circles indicate outliers. The color is associated to the membership to one class and the name of the class is given along the x-axis. The colored areas in the lower right figure are defined by the first quartile and the third quartile of $r$ and $\theta$. Each dot is related to one gauged basin. The doted lines indicate the start of four meteorological seasons.**

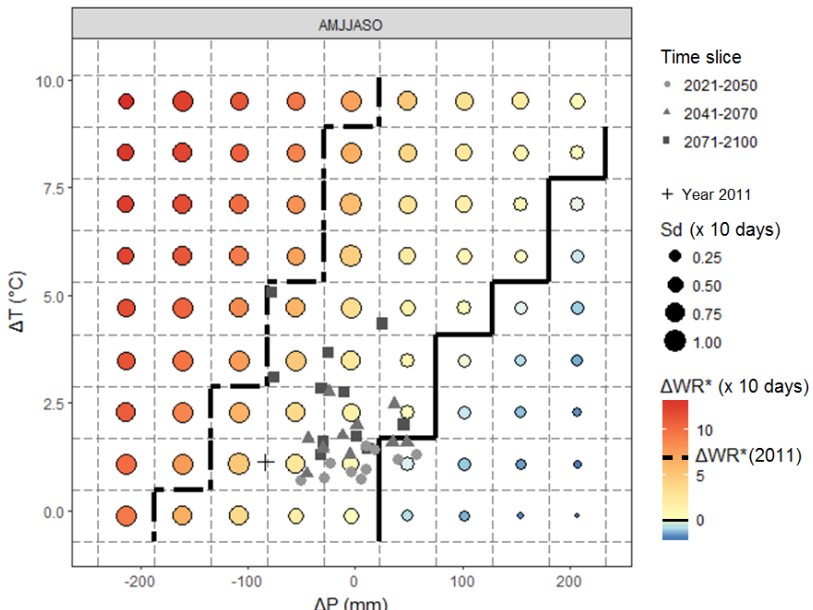


**Figure 12: Climate response surface of legally-binding water restrictions level anomalies ΔWR\* for the Ouche River basin including both exposure and performance characterizations.**



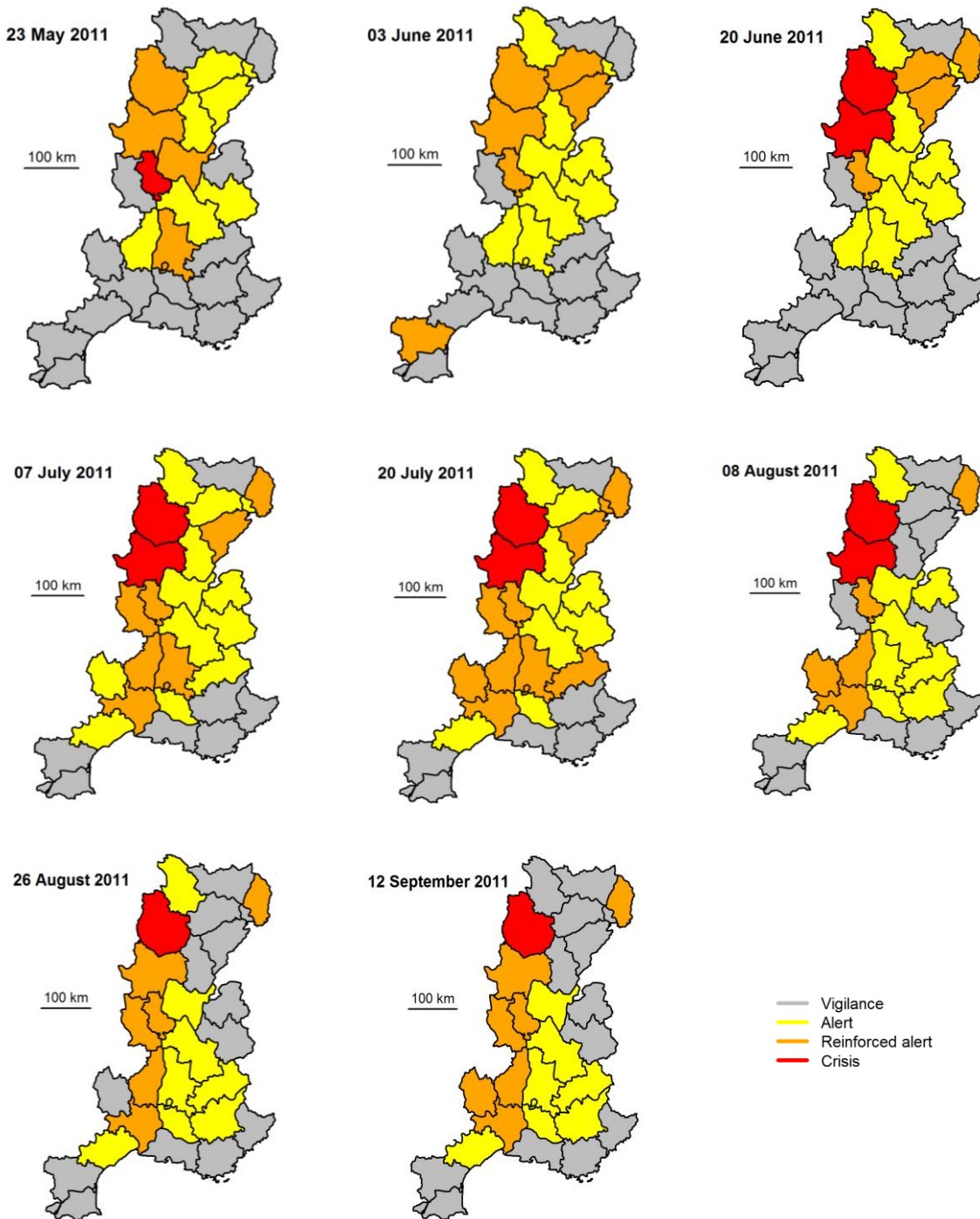

**Figure 13: Most severe water restriction level adopted at the department-level scale for several dates between May and**
**September 2011 (Source: French ministry of Ecology)**

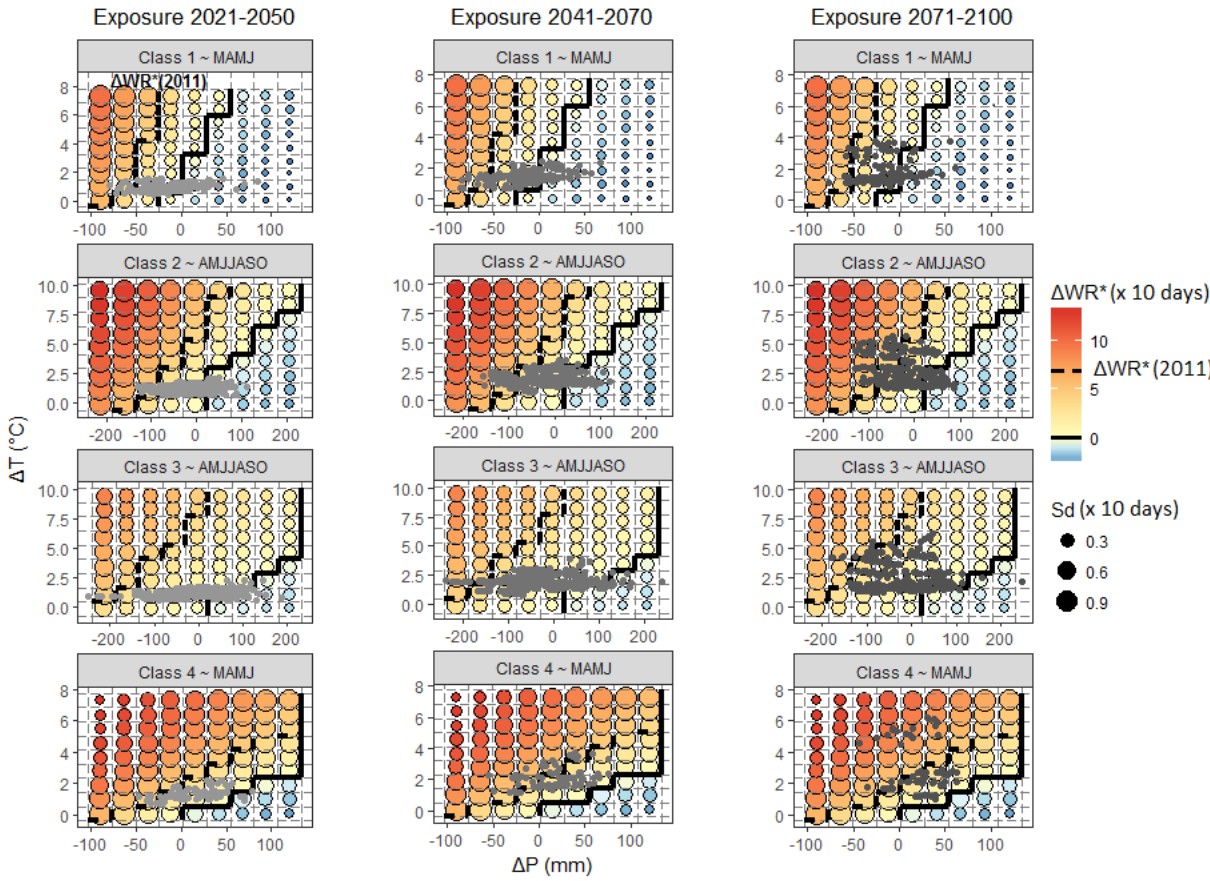


**Figure 14: Representative climate response surfaces for each class including both exposure and performance characterizations.**



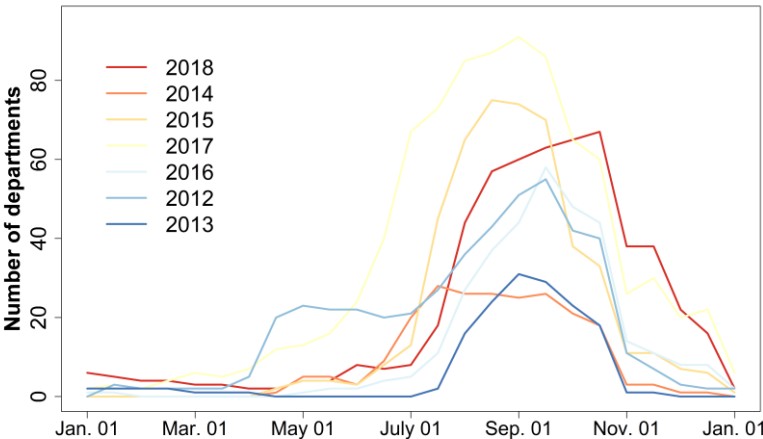


**Figure 15: Number of departments with at least one sub-catchment with WR level ≥ 1. The color of the curves is associated to the annually averaged air temperature rank for France (from red to blue for the warmest (2018) to the coldest year (2013)) (Sources: MétéoFrance, French ministry of Ecology).**



**Appendix A: Classification of river flow regime for France**
Sauquet *et al.* (2008) have defined a classification based on the mean monthly runoff pattern (Fig. A1) and a
map has been published showing the assignment to one class along the main river network. The twelve
dimensionless coefficients *CM* are the twelve values of mean monthly runoff (mm) divided by the mean annual
runoff).
Groups 1 to 6 are pluvial river flow regimes. The six groups mainly differ by the contrast between the maximum
and the minimum of the monthly discharges. Nearly uniform flows through most of the year (Group 1) are found
where large aquifers moderate flows whereas Group 6 is characterized by very low flow in summer, reflecting the
lack of deep groundwater storages in the catchment. Group 7 is representative of Mediterranean river flow regimes
where small rivers basins experience hot and dry summers and intense rainy events in autumn. Their runoff pattern
therefore exhibits severe low flow in summer and high flow in November. In mountainous areas, uppermost basins
display snowmelt-fed regimes (Groups 10, 11 and 12). The lower the outlet is, the lower the contributions of
snowmelt to runoff. Groups 8 to 9 are in the transition regime. The seasonal variation of streamflow is affected as
much by precipitation timing as by air temperature and topographic influences (on snowpack formation and
snowmelt timing). Typically, high flows are observed in spring.

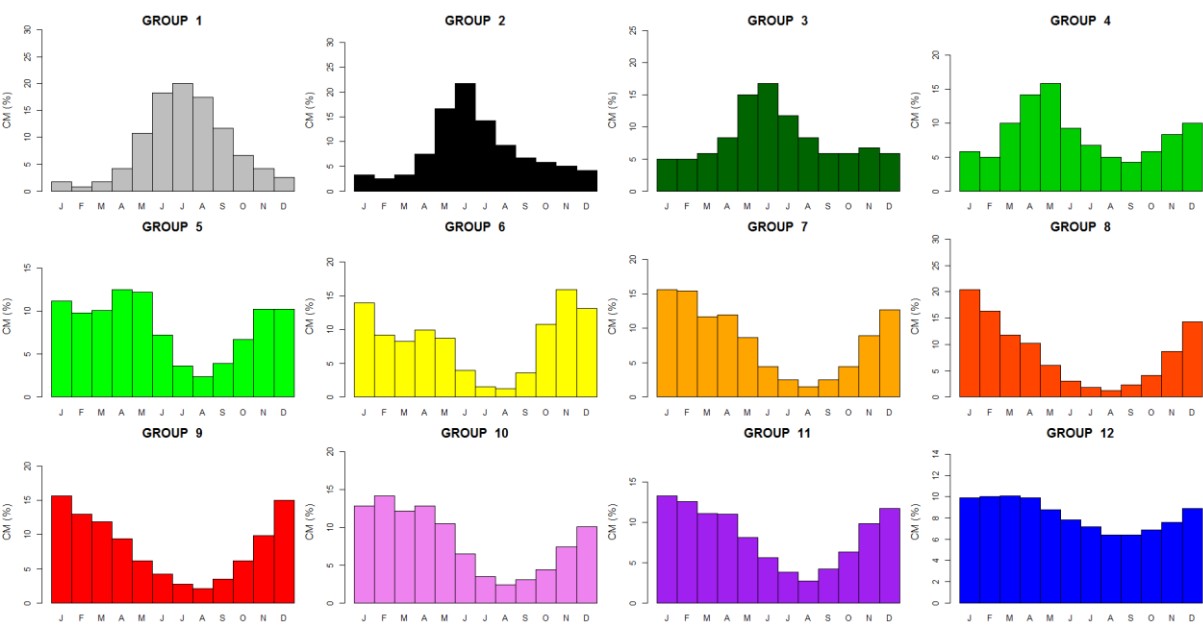


**Figure A1 : Reference dimensionless hydrographs representative of the classification of river flow regime for France**
**(after Sauquet *et al.* 2008)**