# Peer review of "Water restrictions under climate change: a Rhone-Mediterranean perspective combining 'bottom up' and 'top-down' approaches"

_Hydrology and Earth System Sciences, 2018_

## Referee Comment (RC1) · Anonymous Referee #1 · 26 Oct 2018

The paper "Water restrictions under climate change: a Rhone-Mediterranean perspective combining 'bottom-up' and 'topdown' approaches" presents a study that uses decision scaling (Borwn et al., 2012) to evaluate future water restrictions pattern in southeast France. I think the topic fit the scope of this journal, but I have several major concerns on this manuscript.

First, the novelty of the paper is questionable. Applying a bottom-up approach such as decision scaling to evaluate climate change impact uncertainty is not a new topic in the field. Although authors might argue that presenting the climate response surface in WR (not streamflow) is relatively new, I do not see any additional information regarding

policy inform that can be generated from this result. Some visualization techniques used in this paper could be attractive (such as using color to represent mean value and size to represent the s.d.) but I failed to understand the overall scientific contribution of this paper.

Second, the modeling framework is extremely unclear. Authors use Section 4 to explain their method but they spent a lot of space to explain decision scaling which is other people's work. They briefly mention the rainfall-runoff model they used but no details about the actual water restriction level modeling framework (Section 4.3). They explain their concept of computing WR in fair details (which is helpful) but I still do not understand how they build the WRL model. What are the input and output of this model? What parameters can be calibrated in this model? How to authors link this model with the rainfall-runoff model? Information about these is partly provided in Section 4.3 but hard to follow from a reader's perspective.

Third, lack of in-depth discussion on the policy implementation. Given that authors use WR in the climate response surface, one can expect that authors should use a lot of space to link their results to drought policy implementation or some information about the adaptation action. However, only a short discussion of WR has been provided at Section 5.5. Given that this is not a methodological paper, these in-depth discussions become the critical point to prove that this paper is worth to be published because readers around the world can learn from this study and apply it to their own drought management policy.

Finally, the structure of the manuscript and English is extremely difficult for readers to follow. The general outline of the paper follows a typical modeling paper while authors introduce their study area and data than their model. However, as I mentioned above, the modeling framework especially for the "Water restriction level modeling framework" is not clear at all. Also, there are general equations list in the results section (Section 5) and irrelevant results (Line 432-474) presented in the result section. There are A LOT of grammar errors and typos that make the manuscript hard to read. This is surprising

that one of the co-authors is from the UK.

I do not think this draft reach the standard of HESS in its current form. I have several detailed comments below.

Line 34 - What do you mean by "changes" Climate or human activities? Line 35 - What kind of drought? Climatic? Hydrologic? Or economic? Line 86 and 88 - You are arguing with yourself. In Line 86 you said water is abundant globally but Line 88 you said water resources are under high stress. Line 90 - Why 43% is high proportion? It is less than the half. Line 96 - You never explain what is "Drought management plan?" Line 111 and 115 - You are arguing with yourself agian. If water restriction decisions are frequent (Line 115), why these catchments are with minor human influence? Water restriction decisions are human influence. Line 173 to 174 - Will the selection of index affects all of your results? You should discuss this in the discussion section. Line 186 - Why cross a threshold is unsustainable? How do you know it won't come back? Quantify sustainability is a difficult challenge and if you don't know what it is, you should not use the word. Otherwise, you should define sustainability. Line 190 - What do you mean by intersection? Line 215 - I don't see any calculation related to irrigation water use in your 4.3? How you do this? Line 254 - Don't understand what you mean. Line 262 - Why not use the worst WRL as indicators? And also why not just use daily time step as your rainfall-runoff model? Why change it to 10-day? Line 263 to 265 - English is so weird. Line 302 to 303 - I do not understand what is your point here. If you know this, then why don't you model that? This means you understand that just a hydrologic model is not enough to do this type of modeling but you still do it and write a paper about it. This just implies that your model is not only WRONG (as all models are) but also not very USEFUL. Line 357 - What drivers? I thought in climate change studies, T and P changes are drivers. Line 358 to 359 - I don't understand your English. Line 402 - Typo. Line 432 to 474 - I do not understand why you have these results here which are not related to WR. Line 788 - There is no need for Figure 2. Line 797 - The explanation of Figure 5 is unclear. This result in my second major comment regarding

the modeling framework. A better explanation needed. Line 801 - The results are weird here. If your GR67 model is good according to your NSE and Kling–Gupta efficiency, why GR67 and HYDRO show different results in a lot of place in this figure? Does not make sense. Line 819 - If "2" and "3" are similar, why you need to separate them into two categories? Line 822 - The figure at the lower-right corner is unreadable.

---

## Short Comment (SC1) · 2 Nov 2018

This comment was written by a student in the MSc course ESS 401 Current topics in Earth System Science at the University of Zurich, Department of Geography. The students were given the task to select a manuscript in review at one of the EGU journals and to write a review. I discussed this review with the student, and find the comments actually quite valuable. Therefore, I post the review here in the hope editor and authors will find them useful to improve the manuscript.

Best regards, Jan Seibert

[Figure]

In the study of Sauquet et al. the vulnerability of current drought management plans (DMPs) in the Rhône-Méditerranée (RM) are evaluated under future climate. To do so water restrictions (WR) from 2005 and 2016 and hydrological data from 1958 to 2013 were analyzed in 106 catchments to derive a framework to reproduce water restriction durations based on low-flow indicators. As the authors write in this framework socio-political factors that can influence the imposition of water restriction are not included. Based on the drought of 2011 a critical threshold of acceptable WR was defined to decide if the DMPs in the future will still be effective. The study aims to assess the effectiveness of current DMPs under climate change to be able to revise the DMPs for the most vulnerable basins. They find out that in temperature-sensitive catchments the water restrictions will increase significantly in the short term and that for this reason there is a need to adapt the DMPs. In the catchments where the precipitation deter-mines the water restriction, they see difficulties to adapt the DMPS as the uncertainties in precipitation is high. They state in the conclusion section several points they did not include in their study but could play an additional role besides the analyses of water restriction duration influenced by temperature and precipitation. These are for example socio-economic system stressors like agricultural practices, population growth, water demand, etc. which also should be considered in the DMPs.

In my opinion, it is an important topic to discuss the reliability of current decision-making rules regarding water scarcity in the future when climate changes. The method used in this study can give a good overview of where there is a need to rethink the DMPs. But in my opinion, it would be quite important to take the socio-political factors into account in the framework to reproduce water restrictions. A further improvement would be if the economic system stressors would be included to evaluate the DMPs. Therefore the current method has still a lot to improve, and that's why it is not fully clear what the substantial contribution of this paper is. Further, I think the description of the method of the hydrological modeling and the framework to reproduce the water restrictions could be more detailed.

Detailed discussion of the manuscript Major comments:

P1-L22 and P16-L423: The four classes could be explained in P16-L423. The same for Figure 11 and 14, it would be easier to understand if each class would be shortly explained in the figure description.

P2-L54: Is the scenario-neutral approach the same as a bottom-up approach? The authors could use the word "bottom up" as well, as they use it also in the title and it is not used in the rest of the paper. Please clarify difference or similarity.

P4-L106 to P5-L120: In section "2.3 Hydrological data" it would be good if the 15 regimes suggested by Sauquet et al. (2008) could be shortly explained.

P5-L121 to P5-L126: In section "2.4 Climate data" Table 2 the RCP2.6, RCP4.5, RCP8.5 scenarios could be explained. And why is Terray and Boé (2013) not listed there as his projections are used in section "5.1 Definition of perturbed climate conditions to build WR response surfaces"?

P6-L163: Is duration d, the time used for deciding if water restrictions are imposed? In this case, I do not understand what is meant by 10d-VCNd(T) in p7-L171. In Figure 5 VC3 has a value for every day. Is it calculated from the last three days? Please clarify.

P7-L173: VC3 was selected, as it is the most common single indicators used in DMPs of the RM district. I might have missed something, but this seems not to be the case for the 15 test catchments chosen for the evaluation of the WR modeling framework. It is not clear for me how you can compare these different low-flow monitoring indicators with each other. This should be described clearer.

P9-L244: Are the 15 catchments used for calibration or only for evaluation? Please clarify.

P13-L343: It is not clear for me if the perturbation of the climate is based on different climate scenarios as RCP2.6, RCP4.5, RCP8.5 or which exact projection is used. In the reference Terray and Boé, 2013 the authors are using they are also talking of

different projections. This needs to be clarified.

P19-L513 to P19-L518: The first two sentences of the conclusion would better fit in the introduction.

Minor comments: P3-L68: Why not saying Rhone-Méditerranée district in southeastern France to be consistent?

P3-L78 to P4-L95: In section "2.1 Study area" a map or a cross-reference to Figure 1 would help to get an overview of the area.

P4-L90: Why just speaking about the irrigation needs? It might be interesting to get the whole picture for what the water is used.

P4-L106: "2.3 Hydrological data" should be in bold.

P4-L109: I do not understand what the authors mean with "Time series including null values or gaps in the data records above 30% of time were disregarded". Does this mean one null value or 30% null values? Please clarify.

P10-L268: Figure 6 in the figure description: "Table 2" should be "Table 1".

P16-L426: In Table 5 in the table description please add where this standard deviation Sd is taken from.

---

## Referee Comment (RC2) · Anonymous Referee #2 · 15 Nov 2018

Sauquet and colleagues applied a scenario neutral approach to evaluate the implementation of water use restrictions and their impacts on irrigated agriculture. They applied this approach to 15 catchments in the Rhone-Mediterranean region with minimal human influence. Their methods included calibration of a hydrological model to each catchment, sensitivity analyses, assessment of exposure and clustering to identify basins with common characteristics. Strengths of this work include comparison of results regionally and identification of catchment classes, as well as high quality graphics presenting the results. Areas to for improvement include problem framing, the implementation and communication of the sustainability assessment, and explanation of the clustering process and its value. With a clearer problem framing and improved

sustainability assessment I believe the scientific and practical contributions of this work would be clearer. The topic is of interest to HESS readers, and subject to major revision I believe that it would be suitable for publication.

Comments

1. The authors make a strong case for why we care about drought risk under climate change. However, the case for why we need to simulate the implementation of water use restrictions should be stronger. The main question I would like to see the authors address here is: how does the simulation of water use restrictions give us a different picture of impacts or ways to mitigate impacts than simulating streamflow alone?

2. The authors thoroughly review the literature in the scenario neutral and decision scaling methods for assessing climate vulnerability in a bottom-up manner. However, the literature on robust decision making is complementary and should be included in this review. Specifically, there are a few robust decision making studies that assess the performance of existing water management plans [e.g. Lempert and Groves, 2010; Bloom et al., 2013]. The authors should note how their work builds upon or goes beyond these prior works.

3. The sustainability assessment is the key link between the occurrence of water use restrictions and impacts. The authors use critical thresholds as a way to measure sustainability. First, I'm not convinced that is a measure of sustainability. Is it serving as a measure of the sustainability of an agricultural economy? Or something else? Please clarify how it meets a reasonable definition of sustainability. Second, it is not clear how this critical threshold was defined. The authors state that a single critical threshold is applied to all catchments. Is this reasonable given the substantial differences in elevation (and therefore temperatures)? And is the local precipitation factored into this threshold? Lastly, do irrigators or other water users in these catchments have access to other water sources to mitigate impacts (e.g. farm ponds, groundwater)? If so, how does that influence the conclusions?

4. On lines 274 to 275 the authors state that GR6J and HYDRO correctly reproduce water use restrictions but are inconsistent with observation. Do the authors mean that the GR69 and HYDRO produce consistent results, but they are incorrect (i.e. don't match observations)?

5. On line 287 the authors state that the simulated streamflow (from GR6J) produces more accurate water use restriction simulations than the observed streamflow. This strikes me as a case where the model may be right for the wrong reasons – which casts doubt on the later results. How is this counter-intuitive result explained and what are the implications for the interpretation of the results?

6. The authors state that the CART analysis can aid sensitivity assessment at un-modelled catchments. Please address in the conclusions if and how this classification can be helpful for water managers or other scientists.

7. Lastly, there are some typographic errors and awkward phrasing in the manuscript and it would benefit from a thorough review. See a few examples below: a. Line 69 use of word "predisposition" b. Line 402 "thee" should read "three" c. Line 482 "come" should read "some" d. Line 540 use of word "incited"

References

Bloom, E., A. Draper, D. Groves, B. Joyce, M. Rayej, and D. Yates (2013), Evaluating Resource Management Strategies for Update 2013 of the California Water Plan, in World Environmental & Water Resources Congress, pp. 2391–2403.

Lempert, R. J., and D. G. Groves (2010), Identifying and evaluating robust adaptive policy responses to climate change for water management agencies in the American west, Technol. Forecast. Soc. Change, 77(6), 960–974, doi:10.1016/j.techfore.2010.04.007.

---

## Referee Comment (RC3) · Anonymous Referee #3 · 17 Nov 2018

The objective of the study is to develop a risk-based framework to simulate water restrictions (WRs) under climate change in Rhone-Mediterranean district in order to evaluate the vulnerability of current Drought Management Plans (DMPs) to future climate conditions. The proposed framework is based on the assessment of three components: sensitivity of WRs to changes in different climate factors, sustainability of WRs for users and exposure in terms of climate response surfaces.

General comments The paper presents an interesting topic. Although the applied methodology seems appropriate to some extent, it is rather unclear in some parts. Overall, I believe that further details should be added to the paper in order to support

the interpretations and conclusions drawn from the analyses carried out by the authors.

Major comments

Section 3 For the sake of better understanding, I suggest to report the equations of low flow indicators and regulatory thresholds used in the manuscript.

Section 4.2 Details on the rainfall-runoff model should be added, with special reference to the way how the influence of reservoirs is taken into account.

Section 4.3 The description of the water restriction level modelling is unclear in some parts. For instance, I would expect that the comparison between simulated WRLs driven by GR6J data and historical WRs will provide a lower sensitivity score than the comparison with simulated WRLs driven by HYDRO data (considered as benchmark), but it's not (see Lines 287-290). Could it be a consequence of the fact that the model disregards socio-political aspects of the decision making-process?

These sentences must be better explained: lines 295-296- "Furthermore, GR6J performance under low-flow conditions show no statistical link with its WRL modelling performance ..." lines 300-301 "...possible biases in rainfall-runoff modeling does not affect much the ability of the WR modeling framework to simulate correctly or not declared WRs" It seems that despite the difficulties of GR6J model in simulating low-flows accurately, the results of WRL modelling driven by GR6J data are good anyway. How do the authors explain that?

Section 5.3 Vulnerability is computed against a critical threshold. The latter is defined as the difference between the number of WRs simulated by the WR GR6J modelling framework for 2011 and over the baseline period. On the other hand, the Vulnerability Index is computed as the proportion (frequency) of RCM-based simulations that fail above the critical threshold. It sounds like a frequency is compared to a number. I believe that this step must be described in details. For the same reason, it is not clear how the black dotted lines representing the critical threshold are drawn in Figures 10

and 14.

Section 5.4 With regard to the hierarchical cluster analysis for catchment classification at regional scale, the authors should specify the catchment characteristics considered to investigate similarity through the Euclidean distance (see line 421-424). Details on the CART model and its implementation should be added.

Technical comments

In Line 260, "VC3 is with 10d-VCN3(T) each day . . .", something is missing.

In Lines 273-274 "OBS WRLs are correctly reproduced by both GR6J and HYDRO simulations, but also can be consistent with OBS" (???). This sentence is rather misleading, I wonder if "OBS" at the beginning of the sentence could be a mistake and could be deleted.

In line 420: ". . . a classification (of what?) was conducted on to define typical response surfaces, . . .". Please specify.

In line 482: "come catchment" to be replaced by "some catchments".

In line 540: replace "precipitations" with "precipitation".

Missing references:

- Brekke et al., 2009 - Weib, 2011 - Schlef et al., 2018 - Gupta et al., 2009 - Kay et al., 2014

---

## Author Comment (AC2) · 3 Jan 2019

**Water restrictions under climate change: a Rhone-Mediterranean perspective combining 'bottom up' and 'top-down' approaches"**

Sauquet et al.

**J. Seibert jan.seibert@geo.uzh.ch**

5   This comment was written by a student in the MSc course ESS 401 Current topics in Earth System Science at the University of Zurich, Department of Geography. The students were given the task to select a manuscript in review at one of the EGU journals and to write a review. I discussed this review with the student, and find the comments actually quite valuable. Therefore, I post the review here in the hope editor and authors will find them useful to improve the manuscript.

Best regards, Jan Seibert

10

In the study of Sauquet et al. the vulnerability of current drought management plans (DMPs) in the Rhône-Méditerranée (RM) are evaluated under future climate. To do so water restrictions (WR) from 2005 and 2016 and hydrological data from 1958 to 2013 were analyzed in 106 catchments to derive a framework to reproduce water restriction durations based on low-flow indicators. As the authors write in this framework socio-political factors that can influence the imposition of water

15   restriction are not included. Based on the drought of 2011 a critical threshold of acceptable WR was defined to decide if the DMPs in the future will still be effective. The study aims to assess the effectiveness of current DMPs under climate change to be able to revise the DMPs for the most vulnerable basins. They find out that in temperature-sensitive catchments the water restrictions will increase significantly in the short term and that for this reason there is a need to adapt the DMPs. In the catchments where the precipitation determines the water restriction, they see difficulties to adapt the DMPS as the

20   uncertainties in precipitation is high. They state in the conclusion section several points they did not include in their study but could play an additional role besides the analyses of water restriction duration influenced by temperature and precipitation. These are for example socio-economic system stressors like agricultural practices, population growth, water demand, etc. which also should be considered in the DMPs. In my opinion, it is an important topic to discuss the reliability of current decision-making rules regarding water scarcity in the future when climate changes. The method used in this study

25   can give a good overview of where there is a need to rethink the DMPs. But in my opinion, it would be quite important to take the socio-political factors into account in the framework to reproduce water restrictions. A further improvement would be if the economic system stressors would be included to evaluate the DMPs. Therefore the current method has still a lot to improve, and that's why it is not fully clear what the substantial contribution of this paper is.

Further, I think the description of the method of the hydrological modeling and the framework to reproduce the water restrictions could be more detailed.

➔ Authors agree with this remark and the method needs to be more explained. Water restrictions simulations complement studies on the impact of climate change on water resources availability and on water use needs. Indeed water needs can only be met first if water resources are available and second if water abstractions are allowed. Regulatory rules are pieces of the puzzle that should be examined. Roughly speaking studying water restrictions is a way to identify additional future constraints on water users. The regulatory aspects have never been deeply examined in France, perhaps due to the recent implementation of DMPs. This paper presents a first attempt to simulate water restrictions over a large area in France. This paper aims at promoting the approaches developed in parallel by Brown (named 'Decision Tree Framework") and Prudhomme (named "Scenario neutral approach") and one of the challenges was to define critical thresholds of unacceptable number of days with legally-binding WR for irrigation use. This paper suggests using information provided by insurance (here from a national system of compensation) at the regional scale.

*Comments*

P1-L22 and P16-L423: The four classes could be explained in P16-L423. The same for Figure 11 and 14, it would be easier to understand if each class would be shortly explained in the figure description.

➔ Climate response surface of WR* legally-binding water restrictions level anomalies ΔWR* is a graphic representation summarizing the sensitivity of ΔWR* to climatic drivers. They all suggest an increase in the occurrence of legally-binding water restrictions when precipitation decreases or when temperature increases. Additional temperature increase and its associated PET increase can compensate for precipitation increase and lead to decrease in ΔWR*. The response surfaces differ by their flatness (e.g. the response surface of Class 3 displays the less contrasted shape).

P2-L54: Is the scenario-neutral approach the same as a bottom-up approach? The authors could use the word "bottom up" as well, as they use it also in the title and it is not used in the rest of the paper. Please clarify difference or similarity.

➔ According to Culley et al. (2016), "Bottom-up approaches are an alternative to the top-down procedure [...], and have been designed to identify performance thresholds independently from climate models' projections". The approach developed here and based on previous published studies (Prudhomme et al., 2010) does not use downscaled GCMs to describe future climate (scenario-led approaches) but relies on sensitivity-based analyses to a wide range of climate changes, making it scenario-neutral. Ex ante climate projections are considered in the last stages of the procedure to assess the risk of failure. The sentence "specifying relevant critical thresholds is the main task involved in bottom-up approaches" will be added in section 4.1.

P4-L106 to P5-L120: In section "2.3 Hydrological data" it would be good if the 15 regimes suggested by Sauquet et al. (2008) could be shortly explained.

➔ The classification could be given in Appendix.

Sauquet et al. (2008) have defined a classification based on the mean monthly runoff pattern and a map has been published showing the assignment to one class for each basin with drainage area > 50 km².

Groups 1 to 6 are pluvial river flow regimes. The groups mainly differ by the contrast between the maximum and the minimum of monthly streamflow. Nearly uniform flows through most of the year (Group 1) are found where large aquifers moderate flows whereas Group 6 is characterized by very low flow in summer, reflecting the lack of deep groundwater storages in the catchment. Group 7 is representative of Mediterranean river flow regimes where small rivers basins experience hot and dry summers and intense rainy events in autumn. Their runoff pattern therefore exhibits severe low flow in summer and high flow in November. In mountainous areas, uppermost basins display snowmelt-fed regimes (Groups 10, 11 and 12). The lower the outlet is, the lower the contributions of snowmelt to runoff. Groups 8 to 9 are in the transition regime. The seasonal variation of streamflow is affected as much by precipitation timing as by air temperature and topographic influences (on snowpack formation and snowmelt timing). Typically, high flows are observed in spring.

[Figure]

**Reference hydrographs representative of the classification of river flow regime for France (after Sauquet et al., 2008)**

P5-L121 to P5-L126: In section "2.4 Climate data" Table 2 the RCP2.6, RCP4.5, RCP8.5 scenarios could be explained. And why is Terray and Boé (2013) not listed there as his projections are used in section "5.1 Definition of perturbed climate conditions to build WR response surfaces"?

➔ I am not totally sure to understand the question. RCPs are namely "Representative Concentration Pathways" (van Vuuren et al.: The Representative Concentration Pathways: An Overview. Climatic Change, 109 (1-2), 5-31, https://doi.org/10.1007/s10584-011-0148-z, 2011). The study published by Terray and Boé (2013) is based on global climate simulations. This study was used to define the spectrum of changes in temperature and precipitation. Here regional climate projections available in the DRIAS portal are used.

P6-L163: Is duration *d*, the time used for deciding if water restrictions are imposed? In this case, I do not understand what is meant by 10d-VCNd(T) in p7-L171. In Figure 5 VC3 has a value for every day. Is it calculated from the last three days? Please clarify.

5 ➔ The name of the variable is confusing and it will be changed. We suggest modifying the following paragraph to improve the presentation of the WRL modelling framework: "Water restrictions are decided after consulting drought committees that convene irregularly. The time-step for modelling WRL was chosen to be compatible with the frequency of drought committees estimated from the analysis of the water restriction orders: WRL is thus computed at a regular time step of ten days. VC3(*t*) is first computed from daily discharge Q(*t*) every day *t*, WRL(*t*) is then deduced by comparing VC3(*t*) to the

10 four regulatory thresholds and finally a unique representative WR level is assigned to each of the 21 10-day periods defined between the 1$^{st}$ April and the 31$^{st}$ October, as the median of WRL(*t*) observed or simulated within that 10-day period."

P7-L173: VC3 was selected, as it is the most common single indicators used in DMPs of the RM district. I might have missed something, but this seems not to be the case for the 15 test catchments chosen for the evaluation of the WR modeling

15 framework. It is not clear for me how you can compare these different low-flow monitoring indicators with each other. This should be described clearer.

➔ Indeed the decision that lead to selecting VC3 as monitoring variable is was made considering the 28 DMPs and this modality is not prevalent within the 15 test catchments (Figure 3). We will make it clearer in the final version: "*VC*3 was selected as the monitoring indicator and the regulatory thresholds are low flow quantiles 10*d-VCN*3 based on the minimum

20 3-day mean discharges extracted by the block minima approach considering the 37 fixed 10-day time-windows as blocks with return periods, as they are the most common single indicators used in the 28 DMPs of the RM district."

P9-L244: Are the 15 catchments used for calibration or only for evaluation? Please clarify.

➔ They were used both for calibration and for evaluation.

25

P13-L343: It is not clear for me if the perturbation of the climate is based on different climate scenarios as RCP2.6, RCP4.5, RCP8.5 or which exact projection is used. In the reference Terray and Boé, 2013 the authors are using they are also talking of different projections. This needs to be clarified.

➔ The "delta-change" method was used to provide a set of perturbed climates in scenario-neutral approach. Following

30 Prudhomme *et al.* (2010), monthly correction factors $\Delta P$ and $\Delta T$ were considered:

$$\Delta P(i) \ = \ P_0 \ + \ A_P \cdot \cos\left[(i - \varphi_P) \cdot \frac{\pi}{6}\right]. \tag{1}$$

$$\Delta T(i) \ = \ T_0 \ + \ A_T \cdot \cos\left[(i - \varphi_T) \cdot \frac{\pi}{6}\right]. \tag{2}$$

$P_0$ and $T_0 + A_T$ are respectively the mean annual changes in equations (1) and (2), with $i$ referring to month 1 to 12, $\varphi_P$ the phase parameter and $A_p$ the semi-amplitude of change (*e.g.* half the difference between highest and lowest values) in equation (1). The parameters $P_0$, $\varphi_P$, $T_0$ and $\varphi_T$ of single-phase harmonic function were fixed with respect to the range of changes suggested by Terray and Boé (2013). Finally 45 precipitation scenarios were created using 9 values of $P_0$ *i.e.* [-20; -13.3; -6.6; 0; 6.6; 13.3; 20; 26.6; 33.3] mm.an$^{-1}$, by 5 values of *Ap i.e.* [0; 6.6; 13.3; 20; 26.6] mm.season$^{-1}$, while $\varphi_P$ parameter is fixed to 1 to consider minimum change in January and maximum change in July. Likewise, 30 temperature scenarios were set up with 6 values of $T_0$ *i.e.* [0; 1; 2; 3; 4; 5]°C.an$^{-1}$ by 5 values of $A_T$ *i.e.* [-0.5; 0.5; 1.5; 2.5; 3.5]°C.season$^{-1}$ while $\varphi_T$ is fixed to 2°C to get maximum change in August. These details will be given in the next version.

P19-L513 to P19-L518: The first two sentences of the conclusion would better fit in the introduction.

Minor comments:

P3-L68: Why not saying Rhone-Méditerranée district in southeastern France to be consistent?

P3-L78 to P4-L95: In section "2.1 Study area" a map or a cross-reference to Figure 1 would help to get an overview of the area.

P10-L268: Figure 6 in the figure description: "Table 2" should be "Table 1".

P4-L106: "2.3 Hydrological data" should be in bold.

➔ Changes will be made in that sense.

P4-L90: Why just speaking about the irrigation needs? It might be interesting to get the whole picture for what the water is used.

➔ "The total net water withdrawal is around 6 billion of m$^3$ in the period 2008-2013 (water abstraction for cooling nuclear plants and hydropower is excluded) with a high proportion of them to support irrigation needs (3.4 billion of m$^3$, including 2 billion of m$^3$ for channel conveyance). Only 10% of water abstracted for irrigation originate from groundwater. Total annual abstracted volumes for drinking water and for water for industrial uses represent 1.6 and 1 billion of m$^3$, respectively."

P4-L109: I do not understand what the authors mean with "Time series including null values or gaps in the data records above 30% of time were disregarded". Does this mean one null value or 30% null values? Please clarify.

➔ "Time series with more than 30% of missing values or more than 30% of zero flows were disregarded."

P16-L426: In Table 5 in the table description please add where this standard deviation Sd is taken from.

➔ Table 5 is now referred in Section 5.2. "This measure is given by the median and the maximum of *Sd* values of the grid (Table 5)."

⇨ The authors would like to thank Jan Seibert and his students for their helpful comments.

---

## Author Comment (AC3) · 3 Jan 2019

**Water restrictions under climate change: a Rhone-Mediterranean perspective combining 'bottom up' and 'top-down' approaches"**

Sauquet et al.

**Anonymous Referee #2**

5 Sauquet and colleagues applied a scenario neutral approach to evaluate the implementation of water use restrictions and their impacts on irrigated agriculture. They applied this approach to 15 catchments in the Rhone-Mediterranean region with minimal human influence. Their methods included calibration of a hydrological model to each catchment, sensitivity analyses, assessment of exposure and clustering to identify basins with common characteristics. Strengths of this work include comparison of results regionally and identification of catchment classes, as well as high quality graphics presenting

10 the results. Areas to for improvement include problem framing, the implementation and communication of the sustainability assessment, and explanation of the clustering process and its value. With a clearer problem framing and improved sustainability assessment I believe the scientific and practical contributions of this work would be clearer.

➔ Authors agree with this remark and the method (including the definition of sustainability) needs to be more explained.

The topic is of interest to HESS readers, and subject to major revision I believe that it would be suitable for publication.

15

*Comments*

1. The authors make a strong case for why we care about drought risk under climate change. However, the case for why we need to simulate the implementation of water use restrictions should be stronger. The main question I would like to see the authors address here is: how does the simulation of water use restrictions give us a different picture of impacts or ways to

20 mitigate impacts than simulating streamflow alone?

➔ Water restrictions simulations complement studies on the impact of climate change on water resources availability and on water use needs. Indeed water needs can only be met first if water resources are available and second if water abstractions are allowed. Regulatory rules are pieces of the puzzle that should be examined. Roughly speaking studying water restrictions is a way to identify additional future constraints on water users. The regulatory aspects have never been deeply examined in

25 France, perhaps due to the recent implementation of DMPs.

2. The authors thoroughly review the literature in the scenario neutral and decision scaling methods for assessing climate vulnerability in a bottom-up manner. However, the literature on robust decision making is complementary and should be included in this review. Specifically, there are a few robust decision making studies that assess the performance of existing

water management plans [e.g. Lempert and Groves, 2010; Bloom et al., 2013]. The authors should note how their work builds upon or goes beyond these prior works.

➔ Many thanks. The state of the art will be completed with literature on robust decision making.

5  3. The sustainability assessment is the key link between the occurrence of water use restrictions and impacts. The authors use critical thresholds as a way to measure sustainability. First, I'm not convinced that is a measure of sustainability. Is it serving as a measure of the sustainability of an agricultural economy? Or something else? Please clarify how it meets a reasonable definition of sustainability.

➔ Sustainability - like vulnerability - has no universal definition. Sustainability assessment is based on the analysis of failures or
10  unacceptable conditions that lead to low crop yield and quality, and consequently to economic losses at such a level that the national system of compensation is initiated. In this application,

- we assumed that irrigated farming is not sustainable if restrictions during drought periods are , on average, too severe - i.e. duration with limited or suspended abstraction for irrigation above a critical threshold - to ensure enough water for crops;
- since it was not possible to compute the effect of water restrictions on crop yield and quality (no crop modelling was
15  considered here) and on economic losses, we used 'agricultural disaster' notifications as proxies to identify the conditions that would be unacceptable/damaging for farmers activities.

This sustainability is thus indirectly related to agricultural economy (not directly related to losses expressed in euros). We change sustainability for failure analysis in the next version.

20  Second, it is not clear how this critical threshold was defined. The authors state that a single critical threshold is applied to all catchments. Is this reasonable given the substantial differences in elevation (and therefore temperatures)? And is the local precipitation factored into this threshold?

➔ Data are collected by the French ministry of agriculture and they are confidential. The year 2011 was the only year when the national system of compensation has been activated with available data between 1958 and 2013 and the duration of water
25  restrictions were derived individually for each catchment and converted in anomalies $\Delta WR^*(2011)$ with respect to the benchmark value (mean over the period 1958-2013). This dispersion is due to heterogeneity in crops, in irrigation systems, in climate (precipitation, PET, temperature)… at the regional scale leading to locally differentiated sensitivity to water restrictions as well as to biases in WR modelling. Since only the year 2011 it is difficult to conclude on the origin of the dispersion (natural or non-natural). We are convinced that this information is valuable. Finally, simplifying but realistic
30  assumptions are imposed by the lack of detail information; thus only one value was considered despite high dispersion in $\Delta WR^*(2011)$ values (Table 6): the critical threshold was set to the average $\Delta WR^*(2011)$ computed on all catchments of the region under agricultural disaster status in 2011 (6.6 10-day periods), and was used for all classes. Note that this value seems realistic: 6.6 10-day periods = 66 days with restrictions = 30% of the time between between the 1[st] April and the 31[st] October.

Lastly, do irrigators or other water users in these catchments have access to other water sources to mitigate impacts (e.g. farm ponds, groundwater)? If so, how does that influence the conclusions?

➔ More detail will be given on this aspect in section 2.1. In France 80% and 20% of water abstraction are taken from surface water and from groundwater, respectively. In the RM district 10% of water used for irrigation originate from groundwater. Irrigators may have access to small reservoirs (storage capacity usually < 1 Mm$^3$). There is actually a wide discussion about these hydraulic structures in France since their impacts on the ecosystem and their efficiency are not well known (Habets *et al.*: The cumulative impacts of small reservoirs on hydrology: A review. Science of The Total Environment, 643, 850-867, https://doi.org/10.1016/j.scitotenv.2018.06.188, 2018). Most of the small reservoirs are filled by surface water in winter and release water later in summer for irrigation purposes. Water restrictions are not imposed to these reservoirs but we assume here that during severe droughts most of them are empty and thus the influence of auxiliary water sources on the conclusions is limited.

4. On lines 274 to 275 the authors state that GR6J and HYDRO correctly reproduce water use restrictions but are inconsistent with observation. Do the authors mean that the GR6 and HYDRO produce consistent results, but they are incorrect (i.e. don't match observations)?

➔ There is obviously a problem with the phrasing on these lines. "Both GR6J and HYDRO simulations are globally consistent with observed WRLs (OBS). However GR6J and HYDRO results may differ from OBS (e.g. basins 9 to 11 in the Lozère department during the year 2005)."

5. On line 287 the authors state that the simulated streamflow (from GR6J) produces more accurate water use restriction simulations than the observed streamflow. This strikes me as a case where the model may be right for the wrong reasons – which casts doubt on the later results. How is this counter-intuitive result explained and what are the implications for the interpretation of the results?

➔ The discharges simulated by GR6J introduced in the WRL model lead to higher *Sensitivity* scores than those obtained with observed discharges extracted in the HYDRO database. The reasons for this unexpected result have been investigated. In particular we have compared the observed and simulated temporal variability in the time series VCN3. A "smoothing" effect in the GR6J simulations compared to observations was initially suspected. Finally no obvious difference in autocorrelation functions was found between observed and simulated time series. One reason could that the period of interest 2005-2013 – with for some basins only three years with stated water restrictions – may be too short to analyse accurately the relative performance of WRL obtained with OBS and with HYDRO, respectively.

The two scores gives a global insight on the performance of the WRL modelling framework and too much weight should not been given to the differences between scores. In this case, we should conclude that the developed WRL modelling

framework leads to similar results (moderate performance in detecting stated water restrictions during the period 2005-2013) with both data sources HYDRO and GR6J. The WRL modelling framework provides an overview of the on-going drought and the drought committees are partly free to account for this information to state or to postpone water restrictions. The developed framework is a useful tool to predict water restrictions with no interference of lobbies, i.e. only based on the physical processes.

6. The authors state that the CART analysis can aid sensitivity assessment at unmodelled catchments. Please address in the conclusions if and how this classification can be helpful for water managers or other scientists.

➔ The CART algorithm creates the best homogeneous group when splitting the data using through a set of "if-then" logical conditions applied to the most relevant factors, i.e. the decision variables. The result is a decision tree with nodes separating the data into two subgroups. The decision variables known at unmodelled but gauged catchments can be introduced in the chain of rules obtained by CART to finally predict – in this application – the assignment to one of the four classes.

7. Lastly, there are some typographic errors and awkward phrasing in the manuscript and it would benefit from a thorough review. See a few examples below:

a. Line 69 use of word "predisposition" ➔ "The paper develops […] to assess the likelihood of future restrictions depending on their sensitivity, sustainability and exposure to climate deviations"

b. Line 402 "tree" should read "three" ➔ "Finally the vulnerability resulting from the combination of the three components sensitivity, sustainability and exposure"

c. Line 482 "come" should read "some" ➔ "Surprisingly negative values for ΔWR*(2011) are found for  some catchments of Classes 1 and 4"

d. Line 540 use of word "incited" ➔ "Water managers are thus  encouraged"

*References*

Bloom, E., A. Draper, D. Groves, B. Joyce, M. Rayej, and D. Yates (2013), Evaluating Resource Management Strategies for Update 2013 of the California Water Plan, in World Environmental & Water Resources Congress, pp. 2391–2403.

Lempert, R. J., and D. G. Groves (2010), Identifying and evaluating robust adaptive policy responses to climate change for water management agencies in the American west, Technol. Forecast. Soc. Change, 77(6), 960–974, doi:10.1016/j.techfore.2010.04.007.

⇨ The authors would like to thank Reviewer2 for his helpful comments.

---

## Author Comment (AC4) · 3 Jan 2019

**Water restrictions under climate change: a Rhone-Mediterranean perspective combining 'bottom up' and 'top-down' approaches"**

Sauquet et al.

**Anonymous Referee #3**

5   The objective of the study is to develop a risk-based framework to simulate water restrictions (WRs) under climate change in Rhone-Mediterranean district in order to evaluate the vulnerability of current Drought Management Plans (DMPs) to future climate conditions. The proposed framework is based on the assessment of three components: sensitivity of WRs to changes in different climate factors, sustainability of WRs for users and exposure in terms of climate response surfaces. General comments The paper presents an interesting topic. Although the applied methodology seems appropriate to some extent, it is

10   rather unclear in some parts. Overall, I believe that further details should be added to the paper in order to support the interpretations and conclusions drawn from the analyses carried out by the authors.

➔ Authors agree with this remark and the method needs to be more explained.

*Major comments*

15   Section 3 For the sake of better understanding, I suggest to report the equations of low flow indicators and regulatory thresholds used in the manuscript.

➔ Changes will be made in Section 3 to better define the variables of interest:

-   "The low-flow monitoring indicators usually considered are: the daily discharge *Qdaily*, the *d*-day maximum discharge *QCd*, $QCd(t)=\max\left(Qdaily(t'),t'\in[t-d+1,t]\right)$ and the *d*-day mean discharge *VCd*, $VCd(t)=\int_{t-d+1}^{t}Qdaily(t')dt'$ , with

20   duration *d* associated with WR decision varying between 2 and 10 days depending on DMPs."

-   "The threshold associated with WR also varies, generally associated with statistics derived from low-flow frequency analysis, but some being fixed to locally-defined ecological requirements. In the context of DMPs, series of minimum *QCd* or *VCd* are calculated by the block minima approach and thereafter fitted to the lognormal distribution. The block is not the year but the month or given by the division of the year into 10-day time-window. The regulatory thresholds

25   are given by quantiles with four different recurrence intervals associated to the four restriction levels. For example, let us consider thresholds based on the annual monthly minima of *VCNd*. The block minima approach is carried out on the *N* years of records for each month *i*, *i*=1…,12 leading to twelves datasets {*min*{*VCNd(t)*, *month(t)=i*, *year(t)=j*},

*j*=1,…,*N*}. The twelve fitted distribution allows the calculation of 48 values of thresholds (=12 months × 4 levels) with four *T*-year recurrence intervals. To enable comparison of results across all catchments, the same definitions for the monitoring variables and the regulatory thresholds have been adopted for all the catchments. *VC*3 was selected as the monitoring indicator and the regulatory thresholds are low flow quantiles 10*d*-*VCN*3 based on the minimum 3-day mean

5     discharges extracted by the block minima approach considering the fixed 10-day time-windows spanning the year as blocks with return periods, as they are the most common single indicators used in the 28 DMPs of the RM district. Lastly return periods *T* of 2, 5, 10 and 20 years will be associated with the "vigilance", "alert", "reinforced alert" and "crisis" restriction levels, respectively, due to their prevalence in the DMPs"

Section 4.2 Details on the rainfall-runoff model should be added, with special reference to the way how the influence of

10     reservoirs is taken into account.

➔ The GR6J model has six parameters to be fitted (see Figure below): the capacity of soil moisture reservoir (X1) and of the routing reservoir (X3), the time base of a unit hydrograph (X4), two parameters of the groundwater exchange function F (X2 and X5) and a coefficient for emptying exponential store (X6). GR6J is combined with the daily snow module Cemaneige. The catchment is divided into five altitudinal bands of equal area on which snowmelt and snow accumulation processes are

15     represented. For each band, daily meteorological inputs – including solid fractions of precipitation - are extrapolated using elevation as covariate and the snow routine is calculated separately. Finally, its outputs are then aggregated at the catchment scale to feed GR6J. The two parameters of Cemaneige are: the parameter controlling snowpack inertia (X1) and the degree-day coefficient controlling snowmelt (X2). No routine to simulate water management (e.g. reservoir) was considered here since discharges of the 106 gauging stations are weakly altered by human actions or naturalized discharges.

[Figure]

Section 4.3 The description of the water restriction level modelling is unclear in some parts. For instance, I would expect that the comparison between simulated WRLs driven by GR6J data and historical WRs will provide a lower sensitivity score than the comparison with simulated WRLs driven by HYDRO data (considered as benchmark), but it's not (see Lines 287-290). Could it be a consequence of the fact that the model disregards socio-political aspects of the decision making-process?

➔ Inputs of the WRL model are daily discharges and precipitation. Outputs are WRL for each of the 21 10-day periods defined between the 1$^{st}$ April and the 31$^{st}$ October. VC3($t$) is first computed from daily discharge Q($t$) every day $t$, WRL($t$) is then deduced by comparing VC3($t$) to the four regulatory thresholds and finally a unique representative WR level is assigned to each of the 21 10-day periods, as the median of WRL($t$) observed or simulated within that 10-day period. To best match the whole monitoring process stated in most of the DMPs, a simple precipitation correction was applied ("Pcorr", in Fig. 5). It consists to give a 'no alert' when precipitation during the preceding 10 days exceeds 70% of inter-annual precipitation average, regardless of the WR simulation results. The WRL framework is applied to observed and simulated data of both discharge and precipitation. To assess the performance of the WRL model under current condition against stated WR decisions, the WRL model is run with observed daily discharges extracted from the HYDRO database (named "HYDRO" in the text) and with daily discharges simulated by the rainfall-runoff model GR6J forced by the SAFRAN reanalysis (named "GR6J" in the text). In the context of climate change the WRL model is run with daily discharges obtained with GR6J forced by one of the 1350 sets of perturbed precipitation, temperature and PET time series. In this later case the regulatory thresholds are calculated on the simulated discharge time series to limit the possible effect of bias in rainfall-runoff modeling.

➔ The discharges simulated by GR6J introduced in the WRL model lead to higher *Sensitivity* scores than those obtained with observed discharges extracted in the HYDRO database. The reasons for this unexpected result have been investigated. In particular we have compared the observed and simulated temporal variability in the time series VCN3. A "smoothing" effect in the GR6J simulations compared to observations was initially suspected. Finally no obvious difference in autocorrelation functions was found between observed and simulated time series. One reason could that the period of interest 2005-2013 – with for some basins only three years with stated water restrictions – may be too short to analyse accurately the relative performance of WRL obtained with OBS and with HYDRO, respectively.

The two scores gives a global insight on the performance of the WRL modelling framework and too much weight should not been given to the differences between scores. The developed WRL modelling framework leads to similar results (moderate performance in detecting stated legally-binding water restrictions during the period 2005-2013) with both data sources HYDRO and GR6J. The WRL modelling framework provides an overview of the on-going drought and the drought committees are partly free to account for this information, i.e. to state or to postpone water restrictions. The developed framework is a tool to predict water restrictions with no interference of lobbies, i.e. only based on the physical processes.

These sentences must be better explained:

- lines 295-296- "Furthermore, GR6J performance under low-flow conditions show no statistical link with its WRL modelling performance" ➔ "Furthermore, there is no significant link between the GR6J efficiency in simulating low flows ($NSE_{LOG}$) and the performance of the WRL (*Sensitivity* and *Specificity* scores), since the determination coefficients between $NSE_{LOG}$ and *Sensitivity*, and between $NSE_{LOG}$ and *Sensitivity* are lower than 7%."

- lines 300-301 "possible biases in rainfall-runoff modeling does not affect much the ability of the WR modeling framework to simulate correctly declared or not declared WRs" It seems that despite the difficulties of GR6J model in simulating low-flows accurately, the results of WRL modelling driven by GR6J data are good anyway. How do the authors explain that? ➔ The WRL framework is applied to observed and simulated discharge data available before 31st December 2013. In this later case the regulatory thresholds are calculated on the simulated discharge time series to limit the possible effect of bias in rainfall-runoff modelling. The possible reasons of comparable performance between GR6J and OBS is that the WRL framework is carried out using regulatory thresholds derived from GR6J outputs and that even if the discharge data are not exactly reproduced by GR6J, their ranking and their relative position to the regulatory thresholds is correctly reproduced.

Section 5.3 Vulnerability is computed against a critical threshold. The latter is defined as the difference between the number of WRs simulated by the WR GR6J modelling framework for 2011 and over the baseline period. On the other hand, the Vulnerability Index is computed as the proportion (frequency) of RCM-based simulations that fail above the critical threshold. It sounds like a frequency is compared to a number. I believe that this step must be described in details.

➔ Indeed there are two measures of vulnerability. Given one specific climate change projection, a catchment could be judged vulnerable if on average the critical threshold is exceeded. The Vulnerability Index is a proportion reflecting the fraction of RCM leading to critical situations on average. This index is introduced here to account for the uncertainty in climate projections in vulnerability assessment. It should be interpreted as conditional probability (risk) with respect to a set of possible future climates and only used as a relative measure to rank the regions, from the less to the most likely impacted regions.

For the same reason, it is not clear how the black dotted lines representing the critical threshold are drawn in Figures 10 and 14.

➔ The dotted black lines are isopleths connecting points of the response surface with $\Delta WR^* = \Delta WR^*(2011)$. Their location in the response surface depends on the shape of the response surface; this is why the dotted lines differ from one catchment to another in Figure 10, and later from one class to another in Figure 14.

Section 5.4 With regard to the hierarchical cluster analysis for catchment classification at regional scale, the authors should specify the catchment characteristics considered to investigate similarity through the Euclidean distance (see line 421-424). Details on the CART model and its implementation should be added.

➔ CART methods perform successive binary splittings of a given dataset according to decision variables. The algorithm identifies automatically through a set of "if-then" logical conditions the best possible predictors, starting from the most discriminating decision variable to the less important factors, to predict the membership to the one of the four groups. The optimal choices are fixed recursively by increasing the homogeneity within the two resulting clusters. At each step one of the clusters (node) is divided into two nonoverlapping parts.

The list of the potential decision variables by type is:

- Severity:
  - Flow exceeded 95% of the time ($Q95$);
  - Annual minimum 10-day daily mean low flows with a 5-year recurrence interval;
  - Annual maximum deficit below threshold $Q95$ exceeded 20% of time;
- Duration:
  - Annual maximum maximal duration of the continuous sequence of zero flow within the year, exceeded on average every five years ($D80$). Maximum duration of consecutive zero flows ($D$) are sampled by block maxima approach and $D80$ is defined as the empirical 80th percentile of cumulative distribution function of $D$;
  - Seasonal recession time scales ($DT$ and $Drec$). This duration based on the hydrograph defined by the 1-day and 30-day moving average of the 365 long term mean daily discharges, $d= 1,\ldots, 365$ ($Qd$ and $Q30d$,

respectively). *Drec* is defined by the time lapse between the median $Qd50$ and the 90th quantile $Qd90$ of $Qd$ on the falling limb of the hydrograph defined by $Q30d$ and $DT = \ln(Qd50/Qd90)/Drec$;

- Rate of change:
    - Ratio $Q95/Q50$;
    - Concavity index derived from flow duration curve $(Q10 - Q99)/(Q1 - Q99)$ (Sauquet and Catalogne, 2011). This descriptor is a dimensionless measure of the contrast between low-flow and high-flow regimes derived from quantiles of the Flow Duration Curve;
    - Baseflow index (*BFI*). *BFI* is a measure of the proportion of the baseflow component to the total river flow, calculated by the separation algorithm separation suggested by Lyne and Hollick (1979);
    - Class of river flow regime based on average monthly runoff pattern defined by Sauquet *et al.* (2008) (between 1 and 12)
    - Seasonality ratio (*SR*) $SR = Q95_{\text{AMJJASON}}/Q95_{\text{DJFM}}$ ($SR > 1$ for mountainous catchment);
- Frequency:
    - Proportion of years with at least one value $Q < Q95$;
- Timing:
    - Mean day of first occurrence of low flow $< Q95$;
    - Mean and dispersion of the occurrence of flows $< Q95$ within the year ($\theta$ and $r$, $r\sin(\theta)$ and $r\cos(\theta)$. These two variables are circular statistics. Each day $i$ with zero flow is converted into an angular ($ti$) and represented by a unit vector with rectangular coordinates ($\cos(ti)$; $\sin(ti)$). The mean of the cosines and sines defines a representative vector. The value for $\theta$ is obtained by calculating the inverse tangent of the angle of the mean vector and the norm of the mean vector provides a measure of the regularity in the dates (a value close to one indicates a high concentration around $\theta$ while a value close to zero indicates no seasonality).

We will include this list in a table.

*Technical comments*

In Line 260, "VC3 is with 10d-VCN3(T) each day . . .", something is missing.

➔ "VC3 is compared with 10d-VCN3(T) each day . . ."

In lines 273-274 "OBS WRLs are correctly reproduced by both GR6J and HYDRO simulations, but also can be consistent with OBS" (???). This sentence is rather misleading, I wonder if "OBS" at the beginning of the sentence could be a mistake and could be deleted.

➜ There is a problem with the phrasing on these lines. "Both GR6J and HYDRO simulations are globally consistent with observed WRLs (OBS). However GR6J and HYDRO results may differ from OBS (*e.g.* basins 9 to 11 in the Lozère department during the year 2005)."

In line 420: ". . . a classification (of what?) was conducted on to define typical response surfaces, . . .". Please specify.

➜ "a classification of the 106 gauging stations based on the 1350 values of ΔWR* was conducted on to define typical response surfaces"

In line 482: "come catchment" to be replaced by "some catchments".

➜ "are found for  some catchments".

In line 540: replace "precipitations" with "precipitation".

➜ "more accurately temperature and/or precipitation"

*Missing references:*

Brekke et al., 2009: Brekke L.D., Maurer E.P., Anderson J.D., Dettinger M.D., Townsley E.S., Harrison A., and Pruitt T.: Assessing reservoir operations risk under climate change. Water Resour. Res., 45, W04411, doi:10.1029/2008WR006941, 2009.

Gupta et al., 2009: Gupta H. V., Kling H., Yilmaz K., and Martinez G. F.: Decomposition of the mean squared error and NSE performance criteria: Implications for improving hydrological modelling. J. Hydrol., 377, 80–91, https://doi.org/10.1016/j.jhydrol.2009.08.003, 2009.

Kay et al., 2014: Kay A. L., Crooks S. M., and Reynard N. S.: Using response surfaces to estimate impacts of climate change on flood peaks: assessment of uncertainty. Hydrol. Process., 28, 5273–5287, https://doi.org/10.1002/hyp.10000, 2014.

Schlef et al., 2018: Schlef K.E., Steinschneider S., and Brown C.M.: Spatiotemporal Impacts of Climate and Demand on Water Supply in the Apalachicola-Chattahoochee-Flint Basin. J. Water Resour. Plann. Manage., 2018, 144(2): 05017020, 2018.

Weib, 2011: Weiß M.: Future water availability in selected European catchments: a probabilistic assessment of seasonal flows under the IPCC A1B emission scenario using response surfaces. Nat Hazards Earth Syst Sci 11:2163–2171, 2011.

⇨ The authors would like to thank Reviewer3 for his helpful comments.

---

## Author Comment (AC5) · 4 Jan 2019

**Water restrictions under climate change: a Rhone-Mediterranean perspective combining 'bottom up' and 'top-down' approaches"**

Sauquet et al.

**Anonymous Referee #1**

5    The paper "Water restrictions under climate change: a Rhone-Mediterranean perspective combining 'bottom-up' and 'top-down' approaches" presents a study that uses decision scaling (Borwn et al., 2012) to evaluate future water restrictions pattern in southeast France. I think the topic fit the scope of this journal, but I have several major concerns on this manuscript.

First, the novelty of the paper is questionable. Applying a bottom-up approach such as decision scaling to evaluate climate

10    change impact uncertainty is not a new topic in the field. Although authors might argue that presenting the climate response surface in WR (not streamflow) is relatively new, I do not see any additional information regarding policy inform that can be generated from this result. Some visualization techniques used in this paper could be attractive (such as using color to represent mean value and size to represent the s.d.) but I failed to understand the overall scientific contribution of this paper. Second, the modeling framework is extremely unclear. Authors use Section 4 to explain their method but they spent a lot of

15    space to explain decision scaling which is other people's work. They briefly mention the rainfall-runoff model they used but no details about the actual water restriction level modeling framework (Section 4.3). They explain their concept of computing WR in fair details (which is helpful) but I still do not understand how they build the WRL model. What are the input and output of this model? What parameters can be calibrated in this model? How to authors link this model with the rainfall-runoff model? Information about these is partly provided in Section 4.3 but hard to follow from a reader's

20    perspective.

➔ Authors agree with this remark and the method needs to be more explained.

Inputs of the WRL model are daily discharges and precipitation. Outputs are WRL for each of the 21 10-day periods for each year spanning the April-to-October period. $VC3(t)$ is first computed from daily discharge $Q(t)$ every day $t$, $WRL(t)$ is then deduced by comparing $VC3(t)$ to the four regulatory thresholds and finally a unique representative WR level is assigned to

25    each of the 21 10-day periods, as the median of $WRL(t)$ observed or simulated within that 10-day period. To best match the whole monitoring process stated in most of the DMPs, a simple precipitation correction was applied ("Pcorr", in Fig. 5). It consists to give a 'no alert' when precipitation during the preceding 10 days exceeds 70% of inter-annual precipitation average, regardless of the WR simulation results. The WRL framework is applied to observed and simulated data of both discharge and precipitation. To assess the performance of the WRL model under current condition against stated WR

30    decisions, the WRL model is run with observed daily discharges extracted from the HYDRO database (named "HYDRO" in

the text) and with daily discharges simulated by the rainfall-runoff model GR6J forced by the SAFRAN reanalysis (named "GR6J" in the text). In the context of climate change the WRL model is run with daily discharges obtained with GR6J forced by one of the 1350 sets of perturbed precipitation, temperature and PET time series. In this later case the regulatory thresholds are calculated on the simulated discharge time series to limit the possible effect of bias in rainfall-runoff
5    modeling.

Third, lack of in-depth discussion on the policy implementation. Given that authors use WR in the climate response surface, one can expect that authors should use a lot of space to link their results to drought policy implementation or some information about the adaptation action. However, only a short discussion of WR has been provided at Section 5.5. Given
10   that this is not a methodological paper, these in-depth discussions become the critical point to prove that this paper is worth to be published because readers around the world can learn from this study and apply it to their own drought management policy.

➔ Discussing the policy implementation is out of the scope of this article. This paper presents a first attempt to simulate water restrictions over a large area in France. This paper aims at promoting the approaches developed in parallel by Brown
15   (named 'Decision Tree Framework") and Prudhomme (named "Scenario neutral approach") and one of the challenges was to define critical thresholds of unacceptable number of days with legally-binding WR for irrigation use. This paper suggests using information provided by insurance (here from a national system of compensation) at the regional scale.

Finally, the structure of the manuscript and English is extremely difficult for readers to follow. The general outline of the
20   paper follows a typical modeling paper while authors introduce their study area and data than their model. However, as I mentioned above, the modeling framework especially for the "Water restriction level modeling framework" is not clear at all. Also, there are general equations list in the results section (Section 5) and irrelevant results (Line 432-474) presented in the result section.

➔ "Results at both local and regional scales are presented and discussed in Sect. 5 before drawing general conclusions in
25   Sect. 6." ➔ "The scenario-neutral approach is applied at both local and regional scales and results discussed in Sect. 5 before drawing general conclusions in Sect. 6."

There are A LOT of grammar errors and typos that make the manuscript hard to read. This is surprising that one of the co-authors is from the UK.

➔ The text will be screened to correct grammar errors and typos.

30

Line 34 - What do you mean by "changes" Climate or human activities?

➔ We mean both since we are dealing with global change issues.

Line 35 -What kind of drought? Climatic? Hydrologic? Or economic?

➔ "The intensity of the changes is still uncertain, however, climate models agree on significant future increase in frequency and intensity of meteorological, agricultural and hydrological droughts in Southern Europe"

Line 86 and 88 - You are arguing with yourself. In Line 86 you said water is abundant globally but Line 88 you said water resources are under high stress.

➔ At the regional scale, water resources are abundant. However during low flow periods there is an intense competition for water between different users and needs—agricultural, municipal, industrial, the environment—resulting in tradeoffs between human demands and environmental needs. The French RM Water Agency has identified areas with persistent imbalance between water supply and water demand (around 40% of the RM district, http://www.rhone-mediterranee.eaufrance.fr/gestion/gestion-quanti/problematique.php).

Line 90 - Why 43% is high proportion? It is less than the half.

➔ Water abstraction for irrigation needs is ranked first in terms of volume; that is why we have considered that 43 is high proportion. We will make changes in Section 2.1. "The total net water withdrawal is around 6 billion of $m^3$ in the period 2008-2013 (water abstraction for cooling nuclear plants and hydropower is excluded) with a high proportion of them to support irrigation needs (3.4 billion of $m^3$, including 2 billion of $m^3$ for channel conveyance). Total annual abstracted volumes for drinking water and for water for industrial uses represent 1.6 and 1 billion of $m^3$, respectively."

Line 96 - You never explain what is "Drought management plan?"

➔ Drought management plans define specific actions to be undertaken to enhance preparedness and increase resilience to drought. This definition will be included in Section 2.2.

Line 111 and 115 - You are arguing with yourself agian. If water restriction decisions are frequent (Line 115), why these catchments are with minor human influence? Water restriction decisions are human influence.

➔ The 15 catchments are benchmark catchments where near natural drought event can be observed. Water can be taken in another nearby catchment. This will be added in section 2.3 to avoid misinterpretation.

Line 173 to 174 - Will the selection of index affects all of your results? You should discuss this in the discussion section.

➔ Indeed choosing the same definitions for the monitoring indicator and regulatory thresholds may partly explain the deviations to the stated WR. This was a simplification assumption. Before stating for VC3 and 10d-VCN3 the four prevalent modalities have been chosen to implement WR simulations:

| Modality | Monitoring variable | Threshold variable | Benchmark period |
|----------|---------------------|--------------------|------------------|
| MOD1 | QC7 | 10d-VCN3(T) | 1958 – 2013 |
| MOD2 | | m-VCN3(T) | |

| MOD3 | VC3 | 10d-VCN3(T) |
|------|-----|-------------|
| MOD4 |     | m-VCN3(T)   |

where m-VCN3(T) refers to quantiles defined by monthly. Results show a weak sensitivity to the choice of these variables. In terms of sensitivity, MOD3 on average performed best than other modalities for both HYDRO and GR6J simulations on the 15 catchments. MOD3 was finally considered for all the catchments.

[Figure]

**Skill scores obtained for the WR level model over the period 2005-2013. Large dots show the mean values of the skill scores. Colored dotted lines are confidence interval.**

It will be discussed in section 4.3.

10 Line 186 - Why cross a threshold is unsustainable? How do you know it won't come back? Quantify sustainability is a difficult challenge and if you don't know what it is, you should not use the word. Otherwise, you should define sustainability.

➔ Sustainability - like vulnerability - has no universal definition. Sustainability assessment is based on the analysis of failures or unacceptable conditions that lead to low crop yield and quality, and consequently to economic losses at such a level that the

15 national system of compensation is initiated. In this application,

- we assumed that irrigated farming is not sustainable if restrictions during drought periods are , on average, too severe - i.e. duration with limited or suspended abstraction for irrigation above a critical threshold - to ensure enough water for crops;

- since it was not possible to compute the effect of water restrictions on crop yield and quality (no crop modelling was considered here) and on economic losses, we used 'agricultural disaster' notifications as proxies to identify the conditions that

20 would be unacceptable/damaging for farmers activities.

This sustainability is thus indirectly related to agricultural economy (not directly related to losses expressed in euros).

Line 190 - What do you mean by intersection?

➔ Each component contributes to the vulnerability assessment of the system (including its management) to systematic climatic deviation.

Line 215 - I don't see any calculation related to irrigation water use in your 4.3? How you do this?

➔ Only the impact of WR on irrigation has been examined here. Irrigation is selected since it is the sector which consumes most water at the regional scale. Water needs are not computed. The impact of WR is highlighted by the 'agricultural disaster' status notified at the department scale and we have assumed that when the total number of days with legally-binding water restrictions exceeds a fixed threshold (defined using data from the year 2011), the situation for farmers is unacceptable (significant losses) and as a consequence the national system of compensation is initiated.

Line 254 - Don't understand what you mean.

➔ The physical components (drought severity) that lead to WR decisions are only considered in the WRL model and no socio-political factor was taken into account to reproduce water restrictions.

Line 262 - Why not use the worst WRL as indicators? And also why not just use daily time step as your rainfall-runoff model? Why change it to 10-day?

➔ Water restrictions are decided after consulting drought committees that convene irregularly. The time-step for modelling WRL was chosen to be compatible with the frequency of drought committees estimated from the analysis of the water restriction orders: WRL is thus computed at a regular time step of ten days.

Line 302 to 303 - I do not understand what is your point here. If you know this, then why don't you model that? This means you understand that just a hydrologic model is not enough to do this type of modeling but you still do it and write a paper about it. This just implies that your model is not only WRONG (as all models are) but also not very USEFUL.

➔ The model is not totally wrong. The WRL framework is able to reproduce the physical bases in the making-decision process and thus can simulate 68% of the stated restrictions over the period 2005-2013 (performance obtained with "GR6J", section 4.3); reaching 68% is not so bad.

Results of our study (conclusions based on the 15 catchments) show that: (i) surprisingly there are noticeable deviations between the drought severity perceived on discharge data and the final decisions to order restrictions but the decisions are not totally uncorrelated with drought conditions, and (ii) most of the catchments are subject to deviations. The performance is judged acceptable to be applied in the scenario-neutral approach. We are aware that the WRL model is far to be perfect and we are convinced that the WR framework will be improved if relevant socio-economical controls are introduced and it will be certainly a challenging task. Just keep in mind that this study is a first attempt to simulate WR decisions at the regional scale.

Line 357 - What drivers?  I thought in climate change studies, T and P changes are drivers.

➔ Indeed temperature and precipitation are the main physical drivers. Here we wanted to assess if WR is more sensitive to P and T over a specific period.

Line 358 to 359 - I don't understand your English.

➔ Response surfaces have been displayed for different pairs of potential climate drivers (X and Y related to temperature and to precipitation, respectively). Their shapes were first examined (a flat response surface is an evidence of no link between WR and (X, Y)). In addition we have used metrics which measure globally the dispersion around the response surface: the median and the maximum of Sd (small values means small deviation and thus strong links between WR and (X, Y)). The drivers are (X, Y) for which the dispersion is minimal and with the most contrasted surface response.

Line 402 - Typo.

➔ "tree" should read "three": "Finally the vulnerability resulting from the combination of the three components sensitivity, sustainability and exposure"

Line 432 to 474 - I do not understand why you have these results here which are not related to WR.

➔ This section details the result of the classification carried out on the 106 individual WR response surfaces, which is consistent with the title "5.4 Response surface analysis at the regional scale".

Line 788 - There is no need for Figure 2.

➔ Indeed this figure could be deleted.

Line 797 – The explanation of Figure 5 is unclear. This result in my second major comment regarding the modeling framework. A better explanation needed.

➔ Inputs of the WRL model are daily discharges and precipitation. Outputs are WRL for each of the 21 10-day periods for each year spanning the April-to-October period. VC3($t$) is first computed from daily discharge Q($t$) every day $t$, WRL($t$) is then deduced by comparing VC3($t$) to the four regulatory thresholds and finally a unique representative WR level is assigned to each of the 21 10-day periods, as the median of WRL($t$) observed or simulated within that 10-day period. To best match the whole monitoring process stated in most of the DMPs, a simple precipitation correction was applied ("Pcorr", in Fig. 5). It consists to give a 'no alert' when precipitation during the preceding 10 days exceeds 70% of inter-annual precipitation average, regardless of the WR simulation results. The WRL framework is applied to observed and simulated data of both discharge and precipitation. To assess the performance of the WRL model under current condition against stated WR decisions, the WRL model is run with observed daily discharges extracted from the HYDRO database (named "GR6J" in the text) and with daily discharges simulated by the rainfall-runoff model GR6J forced by the SAFRAN reanalysis (named

"GR6J" in the text). In the context of climate change the WRL model is run with daily discharges obtained with GR6J forced by one of the 1350 sets of perturbed precipitation, temperature and PET time series. In this later case the regulatory thresholds are calculated on the simulated discharge time series to limit the possible effect of bias in rainfall-runoff modeling.

Line 801 - The results are weird here. If your GR67 model is good according to your NSE and Kling–Gupta efficiency, why GR67 and HYDRO show different results in a lot of place in this figure? Does not make sense.

➔ The GR6J model is not perfect (both criteria < 1). Small deviations to the observed discharge lead to difference in results obtained by the WR model.

10

Line 819 - If "2" and "3" are similar, why you need to separate them into two categories?

➔ This division into two classes have been suggested by the hierarchical clustering and the response surface representative of Class 2 is more contrasted than that of Class 3.

15    Line 822 - The figure at the lower-right corner is unreadable.

[Figure]

⇨    The authors would like to thank Reviewer1 for his helpful comments.

---

## Referee Report (RR1)

The second version of the manuscript HESS-2018-456 has been significantly improved with respect to several technical issues related to the methodology.

However, I have still some doubts on the WRL modeling results, showing a higher consistency of the hydrological monitoring indicators with the adopted legally-binding water restrictions (WR) when modelled discharges (GR6J) are used as input, rather than when observed discharges (HYDRO) are considered (lines 364-371). Maybe, the calculation of WRLs as the median of the water restriction levels *wrl*(d) for each 10-day period is not a good choice. Have you tried with different statistics, such as the mode? Otherwise, looking at Figure 6, one might think that the hydrological monitoring indicators alone are not enough to explain the reason for the implementation or non-implementation of WRs in some of the investigated catchments in the past years. For instance, negative deviations can derive from an increase in water demands (following section 4.5 changes in water demand are disregarded in this study), whereas positive deviations can be due to the availability of other water sources, such as groundwater or water storage in reservoirs.

With reference to the potential advantage of the proposed approach for decision making in WR policy implementation, and in particular in evaluating the effectiveness of DMPs under climate change and in the definition of priority in reviewing the plans, I believe that, despite the study introduces several simplifications (e.g. WRL modeling neglects: the role of the drought committees and stakeholders, the physical and socio-economic differences among the catchments, the potential changes in water uses and so on), it represents an original and fair attempt to investigate the sensitivity of DMPs to climate change. More specifically, the risk based approach, although improvable in several parts, provides preliminary indications on the vulnerability of irrigation in relation to the likelihood of occurrence and severity of future WRs.

Technical revisions

Lines 68-69: "… in terms of vulnerability to climate change in terms of access to water for agricultural uses." Please rephrase.

Line 173: VCd is defined as a mean discharge, however I think it should be divided by the duration d, otherwise it is a flow volume.

Line 193-195: "Where appropriate, other supporting local observations such as groundwater levels, reservoir water levels, field surveys provided by the ONDE network (Beaufort et al., 2018) or feedbacks from stakeholders can be used to inform final decisions."

Line 231: "In the case of our study, this would be acceptable or not water restrictions for users,". Something is missing in this sentence.

Line 295: what do you mean with "naturalized discharges"?

---

## Author Response (AR2)

**Water restrictions under climate change: a Rhone-Mediterranean perspective combining 'bottom up' and 'top-down' approaches"**

Sauquet et al.

The authors would like to thank the reviewers for their helpful comments. We have introduced new sections: 4.5 to discuss the assumptions consider for water uses and 5.4 to discuss water restriction policy implementation. Additional information is given on the rainfall-runoff model in Section 4.2. Section 3 has been slightly modified also (discussion on DMPs and on hydrological indices).

[revised manuscript text omitted]

---

## Author Response (AR3)

**Water restrictions under climate change: a Rhone-Mediterranean perspective combining 'bottom up' and 'top-down' approaches"**

Sauquet et al.

**Anonymous Referee #2**

The second version of the manuscript HESS-2018-456 has been significantly improved with respect to several technical issues related to the methodology.

However, I have still some doubts on the WRL modeling results, showing a higher consistency of the hydrological monitoring indicators with the adopted legally-binding water restrictions (WR) when modelled discharges (GR6J) are used as input, rather than when observed discharges (HYDRO) are considered (lines 364-371). Maybe, the calculation of WRLs as the median of the water restriction levels wrl(d) for each 10-day period is not a good choice. Have you tried with different statistics, such as the mode? Otherwise, looking at Figure 6, one might think that the hydrological monitoring indicators alone are not enough to explain the reason for the implementation or non-implementation of WRs in some of the investigated catchments in the past years. For instance, negative deviations can derive from an increase in water demands (following section 4.5 changes in water demand are disregarded in this study), whereas positive deviations can be due to the availability of other water sources, such as groundwater or water storage in reservoirs.

➔ A discussion was introduced specifically on this aspect in Section 4.3. "*Heterogeneity in basin characteristics and rules imposed by the DMPs should not result in a systematic difference in Sensitivity and Specificity score between GR6J and HYDRO identified for most of the 15 evaluation catchments. Simulations were made on near pristine catchments and thus water uses are unlikely to be the main reason. Other causes of higher Sensitivity scores obtained when simulated discharges are used as input have been investigated in the WRL modeling framework. However, results of this analysis have not been conclusive. The aforementioned tests with the four prevalent modalities have all led to higher Sensitivity score using GR6J and higher Specificity score using HYDRO, demonstrating that the choice of the monitoring indicator and regulatory thresholds is probably not involved. A "smoothing" introduced by the hydrological modelling was also suspected but autocorrelation in observed and GR6J simulated VC3 time series was found very similar. Future works may re-investigate these aspects. They will need to explore new ones (e.g., the way WRL is derived from the daily values wrl for each 10-day period) using a longer verification period with not necessary uniform but fixed regulatory framework. Indeed some catchments have experienced only three years with legally-binding water restrictions and DMP have been frequently during the 2005-2013 period (see the black vertical segments in Fig. 6).*"

A new comment was added on the comparison between results obtained by HYDRO and GR6J. "*Using GR6J is more effective for detecting legally-binding restriction than using observed discharges while it is less efficient for predicting periods without restriction for most of the catchments. There is a compensatory effect, which is not easy to detect graphically since Sensitivity scores are more sensitive than Specificity scores due to the reduced number of observed days with adopted restrictions.*"

*Technical revisions*

Lines 68-69: "… in terms of vulnerability to climate change in terms of access to water for agricultural uses." Please rephrase.

➔ The sentence has been modified: "*aims to establish a ranking of areas vulnerable to climate change in terms of water access for agricultural uses.*"

Line 173: *VCd* is defined as a mean discharge, however I think it should be divided by the duration *d*, otherwise it
is a flow volume. ➜ The definition of the mean discharge *VCd* has been modified: $VCd(t)=\frac{1}{d}\int_{t-d+1}^{t}Qdaily(t')dt'$

Line 193-195: "Where appropriate, other supporting local observations such as groundwater levels, reservoir water
levels, field surveys provided by the ONDE network (Beaufort et al., 2018) or feedbacks from stakeholders can be
used to inform final decisions." ➜ The sentence has been modified.

Line 231: "In the case of our study, this would be acceptable or not water restrictions for users,". Something is
missing in this sentence. ➜ The sentence has been clarified: *"In the case of our study, these thresholds will make*
*it possible to distinguish duration of water restrictions, which are unacceptable for users."*

[revised manuscript text omitted]